# Next-Gen intestinal parasite detection: Leveraging metataxonomics for improved diagnosis of intestinal protists and helminths

Katherine Bedoya-Urrego[1], Nicolas Rozo-Montoya[1], Ana L. Galván-Díaz[4], Gisela M. Garcia-Montoya[1,2,3], Juan F. Alzate [1,2,3]*

**1** Centro Nacional de Secuenciación Genómica - CNSG, Sede de Investigación Universitaria- SIU, Universidad de Antioquia UdeA, Calle 70 No. 52-21, Medellín, Colombia, **2** Departamento de Microbiología y Parasitología, Facultad de Medicina, Universidad de Antioquia UdeA, Calle 70 No. 52-21, Medellín, Colombia, **3** Grupo Pediaciencias, Facultad de Medicina, Universidad de Antioquia UdeA, Calle 70 No. 52-21, Medellín, Colombia, **4** Grupo de Microbiología ambiental, Escuela de Microbiología, Universidad de Antioquia UdeA, Calle 70 No. 52-21, Medellín, Colombia

* jfernando.alzate@udea.edu.co

## Abstract

Intestinal parasites continue to pose a significant public health burden in low- and middle-income countries and are increasingly recognized in developed regions. Traditional diagnostic methods, primarily based on microscopy, remain widely used despite limitations in sensitivity and taxonomic resolution. In this exploratory study, we applied a next-generation sequencing (NGS)-based metataxonomic approach, integrated with classical phylogenetic methods, to characterize intestinal parasites in rural Colombian populations. We compare its performance with conventional microscopy, focusing on both protist and geohelminth detection. Metataxonomics outperformed microscopy in detecting *Strongyloides stercoralis* and enabled precise species and subtype level assignment for *Blastocystis* and *Entamoeba* spp., revealing frequent mixed infections. Microscopy detected *Trichuris trichiura*, *Giardia*, *Cyclospora*, and *Chilomastix* more effectively, highlighting limitations of current primers and DNA extraction methods. *Cystoisospora* was only identified by NGS. These results demonstrate the utility of NGS-based metataxonomics for broad parasite detection while emphasizing areas for methodological improvement and providing a foundation for future, larger-scale studies.

## Introduction

Intestinal protist and helminth infections remain a significant public health challenge globally, particularly in developing countries, where poor sanitation, and limited access to safe drinking water, inadequate diagnostic capacity, and widespread socioeconomic vulnerability contribute to the transmission [1]. These conditions result in persistent fecal contamination of soil, water, and food, facilitating the continued

**Data availability statement:** Data are available at NCBI SRA: https://www.ncbi.nlm.nih.gov/bioproject/PRJNA1293464.

**Funding:** This study was funded by Escuela de Microbiología-CODI, Universidad de Antioquia, under grant code 2023-64370. The funders had no role in study design, data collection and analysis, decision to publish, or preparation of the manuscript.

**Competing interests:** NO authors have competing interests.

spread of these infections. Although governmental initiatives, often supported by the World Health Organization (WHO), have been implemented to curb transmission, actions focused on health promotion and infection prevention and control remain insufficient and largely ineffective.

Globally, approximately 3.5 billion people are infected with intestinal protozoa and helminths, of whom around 450 million are symptomatic, leading to significant morbidity and mortality [2]. These parasites contribute substantially to gastrointestinal disorders such as diarrhea, chronic malabsorption, malnutrition, and impaired growth and development [3]. The associated health consequences are particularly severe among vulnerable populations, including school-age children, pregnant women, and immunocompromised individuals. The resulting burden highlights the need for improved detection and control strategies.

Among intestinal protozoan parasites, *Giardia intestinalis*, *Entamoeba histolytica*, *Cryptosporidium* spp. and *Blastocystis* spp. are the most reported worldwide [4]. In parallel, soil-transmitted helminths (STH) affect more than 1.5 billion people globally, accounting for nearly 24% of the world's population [5]. The most frequently reported species include *Ascaris lumbricoides*, *Trichuris trichiura*, and the hookworm *Ancylostoma duodenale* and *Necator americanus* [6,7]. Additionally, the threadworm *Strongyloides stercoralis*, while significant and likely underreported, is often considered separately [6]. Zoonotic STH species such as *Ancylostoma ceylanicum*, *Ascaris suum*, *Trichuris vulpis*, and *Trichuris suis* have also increasingly been detected in humans, expanding the spectrum of helminths known to cause disease in the human host [8].

Microscopy-based methods are the common standard for intestinal parasite detection due to their low cost and simplicity [9]. However, they exhibit low sensitivity and specificity which is highly dependent on the parasite load and technician expertise [10]. They also lack the resolution needed to accurately differentiate morphologically similar species, such as *Entamoeba histolytica* from non-pathogenic *Entamoeba* spp., *A. lumbricoides* from closely related zoonotic species like *A. suum*, or among human-infecting hookworms such as *N. americanus* and *A. duodenale* [10,11]. Consequently, there is a growing need for studies using advanced diagnostic tools to reliably identify parasite species infecting human and animal populations in specific areas [12–14].

To overcome these limitations, PCR-based methods have been widely adopted, offering superior sensitivity and specificity for detecting specific parasite species or genera [13,15]. A significant advance has been the development of multiplex PCRs for the simultaneous detection of several parasites in a single reaction [13,16–19]. However, these molecular approaches present their own challenges. Current multiplex PCR assays for intestinal parasites exhibit significant limitations in their target selection, creating diagnostics gaps that may compromise comprehensive parasitological assessment. These assays are specifically designed to detect established pathogenic species, most of them protozoa, systematically excluding helminths and non-pathogenic organisms.

Next-generation sequencing (NGS) strategies emerge as a powerful alternative, enhancing taxonomic resolution through untargeted detection and comprehensive

characterization of parasite diversity. These approaches overcome the limitations of specific PCR methods by enabling the simultaneous identification of multiple taxonomic groups, either via direct DNA sequencing (metagenomic) or PCR-based amplification of specific genetic markers (metataxonomic) [20–22]. This latter strategy has become invaluable in microbial ecology due to their cost-effectiveness and capacity to identify a broad range of organisms within a single sample. Metataxonomic analyses typically target ribosomal RNA gene regions, such as 16S for bacteria, 18S for eukaryotes, and the internal transcribed spacer (ITS) region [23]. While widely applied in bacterial research, amplicon-based analysis in parasitology remains limited. Applications in protists have shown promising results [24–29], yet its implementation for human associated helminth detection is still scarce [30]. In protist research, it has revealed greater diversity than traditional methods, enabling the identification of mixed infections and novel genetic variants of parasites like *Blastocystis* [24,31–33], as well as the detection of various intestinal protists (*Entamoeba*, *Iodamoeba*, *Balantioides*, *Tetrathrichomonas*) and taxa typically excluded from routine diagnostics, such as *Dientamoeba fragilis* [22,34]. In helminths, NGS has been validated for taxonomic classification, demonstrating efficacy in diverse contexts from soil nematode biodiversity to the identification of key soil-transmitted helminths (STH) including *Ascaris*, *Trichuris*, and *Strongyloides* in modern and archaeological samples [35–38], as well as trematodes such as *Schistosoma* spp. and cestodes like *Taenia* spp. and *Echinococcus* spp. [39]. Together, these findings demonstrate that sequencing-based approaches not only mitigate the limitations of traditional methods but also opens new pathways for the discovery of unexpected pathogens.

In response to the recognized diagnostic gaps in parasitology, this study explores the application of a metataxonomic approach coupled with phylogenetic methods for the simultaneous detection and, when possible, species-level confirmation of multiple parasite taxa in human fecal samples from Colombia. By evaluating a scalable, high-throughput strategy, the study aims to contribute to ongoing surveillance efforts and support more accurate monitoring in endemic settings.

## Materials and methods

### Sample description and microscopic diagnosis

A total of 65 fecal samples were selected in the present study based on confirmed microscopic diagnoses of the soil transmitted helminths and protists. Fecal samples were obtained from individuals in selected regions of Colombia as part of previous research projects. At the time of collection, stool samples were processed using direct saline and iodine smears, following a concentration step using the MiniParasep® SF fecal parasite concentrator (Apacor Ltd, England). The microscopic examination was conducted by an experienced staff of professionals with specific expertise in intestinal parasite diagnosis, following standardized internal protocols aligned with clinical laboratory guidelines [9]. To ensure maximum sensitivity, the entire surface area of each prepared slide (both direct and concentrated) was systematically examined under 100x and 400x magnification. The competence of the technical staff was rigorously maintained through a continuous quality assurance system that included periodic validation of diagnostic skills using reference slide collections. Additionally, a senior parasitologist was available for case consultation and verification of uncertain morphological identifications. This approach ensured standardized morphological identification and consistent performance across all operators. Samples were stored at -80ºC until DNA extraction. Prior to molecular analysis, all samples were anonymized and coded to ensure the confidentiality of participants. A complete description of the parasitological results is presented in S1 Table.

### DNA extraction and metataxonomic experiment

DNA extraction was performed with the Stool DNA isolation kit NORGEN BIOTEK CORP. DNA quantitation and UV spectral analysis was performed with Nanodrop 2000c to assess DNA purity and quantity. The V4 region of the eukaryotic rDNA gene was selected as marker for this study using the primer pair: 18S-V4Fw: CCAGCAGCCGCGGTAATTCC [40], and the reverse primer was 18S–V4 Rev: RCYTTCGYYCTTGATTRA, as described elsewhere [41]. Amplicon libraries were prepared and sequenced at Macrogen (Seoul, Korea) in a MiSeq (Illumina) platform to generate paired end reads of 300 bases.

## Bioinformatic processing

Sequencing depth and quality were evaluated across all samples to ensure adequate coverage for metataxonomic analyses. The number of raw read pairs per sample ranged from **64,254–214,280**, with post-filtering Q30-cleaned read pairs ranging from **31,891–120,273**. Read cleaning was performed with cutadapt v2.10. After applying the Mothur quality-control pipeline, the number of high-quality sequences retained per sample ranged from **5,884–109,502**. The estimated Good's coverage values varied between **0.996019 and 0.99997**, indicating that the sequencing effort was sufficient to capture nearly the entire diversity of dominant taxa present in each sample (S2 Table).

Amplicon reads underwent processing using the MOTHUR pipeline version 1.44 [42]. Briefly, Paired-endPaired end (PE) reads were merged using Mothur's command "make.contigs". Sequences with homopolymers longer than 8, with ambiguous bases, or sequences shorter than 250 bases were filtered out. The chimeric sequences were detected with VSEARCH [43] and removed from the analysis. Read clustering to operational taxonomic units (mOTUs) was performed with the subroutine "dist.seqs" at 97% nucleotide identity. Library size for each sample was normalized with the "totalgroup" method.

The taxonomic assignment of the generated mOTUs was conducted by comparisons with the SILVA v138 ribosomal database [44] in the MOTHUR pipeline. mOTUs identified as protist or nematode were retained for subsequent alignment with the BLASTN tool [45]. We selected good quality sequences with a length ≥1000 bp and considered valid candidate mOTUs for subsequent phylogenetic analyses those that showed a bit score ≥400 in the BLASTN.

To perform the taxonomic assignation, a database was built with mOTUS harbouring sequences of the of interest present in the stool samples and a curated reference sequence database of 18S rDNA genes of the nematode and protists. The reference protist included species of the genus *Entamoeba, Endolimax, Iodamoeba, Cystoisospora, Besnoitia, Hyaloklossia, Caryospora, Eumonospora, Hammondia, Neospora, Toxoplasma, Nephroisospora* and *Blastocystis* subtypes (ST1 to ST17). We selected nematodes in the superfamily Ancylostomatidae (*Rhabditella axei, Angiostrongylus cantonensis, Angiostrongylus costaricensis, Angiostrongylus vasorum, Haemonchus contortus, Ancylostoma duodenale* and *Necator americanus)*, Dorylaimia subclass (*Soboliphyme baturini, Trichuris muris, Trichuris suis, Trichuris trichiura, Trichinella spiralis, Trichinella pseudospiralis),* Strongyloididae family (*Strongyloides papillosus, Strongyloides ransomi, Strongyloides ratti* and *Strongyloides stercoralis),* Spirurina (*Toxocara vitulorum, Toxocara canis, Toxocara cati, Ascaris lumbricoides,* and *Ascaris suum).* The accession number for the reference sequences can be found in S3 Table.

The reference sequences of each taxonomic group, along with the mOTUs identified by BLASTN comparisons as candidate helminth and protists species were aligned with MAFFT v7.215 software [46]. The phylogenetic tree was calculated using the IQ-TREE2 software [47] with 1000 ultrafast bootstrap pseudoreplicates to test the topology of the trees [48]. The ModelFinder was used to automatically select the best substitution model for each taxa dataset [49].

After the confirmation of the taxonomic assignment for each protist and nematode mOTU, a database was elaborated to summarize the findings from both the microscopic smear analysis and the NGS metataxonomic analysis. The presence of a classified mOTU in any individual was noted as a qualitative value of "present". Individuals in which the nematode mOTU was absent were categorized as "negative."

## Statistical analysis

Descriptive analyses were performed to compare the detection outcomes obtained by microscopy and metataxonomic (NGS-based) approaches. The presence or absence of each parasite taxon was summarized as qualitative variables, and detection frequencies were calculated as the proportion of positive samples per method. Simple concordance rates between both diagnostic approaches were also calculated to provide a preliminary measure agreement. Results were visualized using comparative bar plots generated in R (v4.3.1) with the ggplot2 package. No inferential statistical tests were applied, as the objective of this preliminary study was to provide an exploratory, qualitative comparison of diagnostic performance. More comprehensive statistical analyses could be performed in future studies with larger sample sizes.

## Ethical considerations

Fecal samples were collected in two time periods: April-November 2015 and September 2022-February 2023 as a part of independent research projects. The use of archived, anonymized fecal samples in this study was approved by the Bio-ethics Committee of Sede de Investigación Universitaria- Universidad de Antioquia (approval code CBE SIU 250, Date: October 18, 2023), in accordance with institutional and national guidelines. The authors had access to epidemiological data from the population; however, all information was anonymized, and participants provided written informed consent prior to data collection. No new samples were collected, and informed consent was not required due to the retrospective and anonymized nature of the material.

## Results

### Parasitic diagnosis using next-generation sequencing and direct examination

This study aimed to provide a preliminary comparison of parasite detection in human fecal samples using two approaches: traditional smear microscopy performed by trained parasitologists, and a metataxonomic method targeting the V4 hyper-variable region of the eukaryotic 18S rDNA. Previous tests confirmed that the primers used in this study are effective for detecting protists and nematodes but do not amplify DNA from cestodes and trematodes. As a result, these groups were excluded from the scope of our analysis. Accordingly, our study focused on the detection of human intestinal protists and nematodes.

The study population was selected due to the high prevalence of intestinal parasitic infections in the region, particularly geohelminths. This selection enabled a robust comparison between the performance of traditional microscopy and metataxonomic (NGS-based) approaches. We began by focusing on nematode infections. Three samples that tested negative for nematodes by microscopy were included as controls. Notably, one of these was negative for all parasites by both methods—NGS and microscopy—serving as a negative reference.

Based on microscopy results, 95.4% of individuals (62 out of 65) were positive for at least one nematode species. In comparison, the metataxonomic approach using NGS detected nematodes in 86.2% of individuals (56 out of 65). As expected, the geohelminths *Trichuris*, *Ascaris*, and hookworms were the most prevalent parasites detected. Notably, hookworms surpassed *Ascaris* in prevalence, displacing it from its typical second position in Colombia. Interestingly, *Strongyloides* was also detected, despite being less commonly reported. When comparing the performance of the two detection methods, microscopy outperformed the metataxonomic NGS approach in identifying *Ascaris* by one additional positive sample and *Trichuris* by twelve. In contrast, both methods yielded the same results for hookworm detection. Strikingly, NGS substantially outperformed microscopy in the detection of *Strongyloides*, identifying three positive samples compared to just one by microscopy—representing a 300% increase (Fig 1).

### Phylogenetic species analysis of nematodes

To improve the taxonomic resolution of nematode identification, we performed phylogenetic analyses using the molecular operational taxonomic units (mOTU) sequences. Curated reference sequences for each of the four detected nematode genera were included to ensure accurate and reliable taxonomic assignment.

*Trichuris* mOTUs were confirmed as *T. trichiura* based on phylogenetic analysis, as they formed a well-supported monophyletic clade with curated reference sequences of *T. trichiura*, receiving a UFB (ultrafast bootstrap) support value of 96 (Fig 2).

The phylogenetic analysis of mOTUs assigned to hookworms successfully identified them as *N. americanus*. The resulting phylogenetic tree showed a well-supported clade containing the human-infecting genera *Ancylostoma* and *Necator*. Reference sequences of *Ancylostoma* formed a distinct branch with 100% ultrafast bootstrap (UFB) support; however, none of the mOTUs from our samples clustered within this branch. Instead, all mOTUs grouped with *N. americanus*

## Comparison of Nematode Detection: NGS vs Microscopy

**Fig 1. Comparison of nematode detection frequencies between NGS-Metataxonomics and microscopy-based diagnostics.** Bar plots show the number of positive samples for four relevant geohelminths—*Ascaris*, *Necator*, *Strongyloides*, and *Trichuris*—detected using either molecular (NGS) or traditional microscopy approaches (Mic). Detection was binarized (positive vs. negative) and frequencies (Y-axis) were calculated across all samples in the dataset. Bars are grouped by parasite and color-coded by detection method (dark blue for NGS and light blue for microscopy).

reference sequences, forming a clade with 87% UFB support. Based on this phylogenetic placement, we assigned the detected mOTUs to *N. americanus*. In conclusion, all individuals in this study who tested positive for hookworms were infected only with *N. americanus*. (S1 Fig).

Similarly, *Ascaris* mOTUs were confirmed to belong to the genus *Ascaris*, forming a clade with a moderate ultrafast bootstrap (UFB) support value of 92. However, species-level assignment was not possible, as the 18S rDNA sequences of the closely related species *A. lumbricoides* and *A. suum* are highly similar, making it impossible to resolve them using this molecular marker (Fig 3).

Finally, the *Strongyloides* sequences from the three infected individuals clustered into a single mOTU, which was assigned to *S. stercoralis*. This mOTU formed a well-supported clade with 100% ultrafast bootstrap (UFB) support along-side curated reference sequences of *S. stercoralis*, confirming its taxonomic identity. Interestingly, one sample was positive for *S. stercoralis* by microscopy, based on the visualization of larvae, but was classified as negative for this species by NGS. Instead, NGS identified this sample as positive for *N. americanus*. Remarkably, all three *S. stercoralis* infections were exclusively detected by NGS, highlighting its enhanced sensitivity for detecting this parasite (Fig 4).

Based solely on the NGS results—given their higher specificity—we found that 32.1% of individuals harbored single nematode infections, while 67.9% presented with mixed infections. The most common coinfection was *N. americanus + T.*

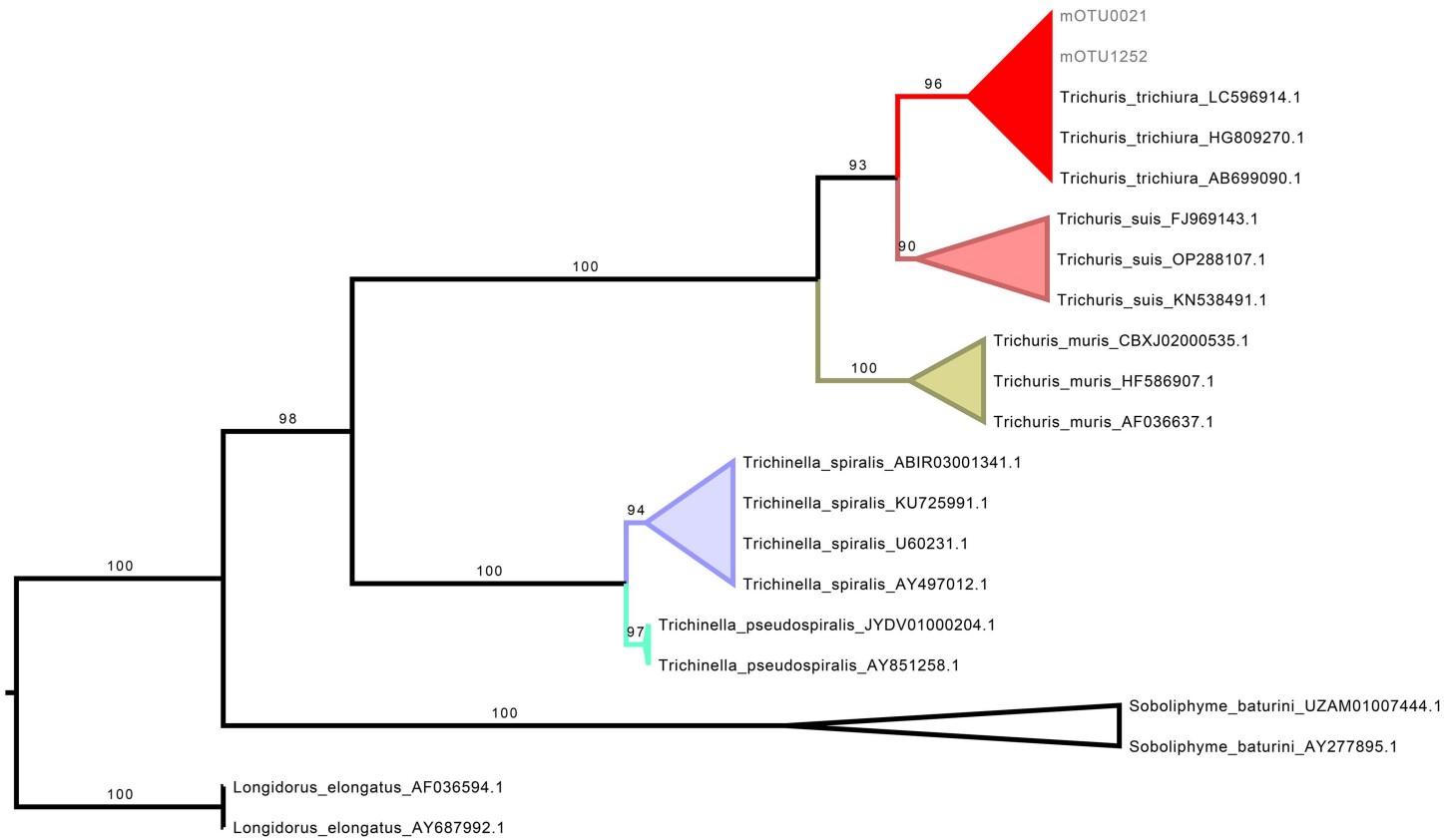

**Fig 2. Phylogenetic analysis of *Trichuris* mOTUs for taxonomic assignment.** Maximum-likelihood phylogenetic tree based on the SSU rRNA gene, constructed using a curated selection of Dorylaimida nematode sequences, including representatives of the genus *Trichuris*. The tree was inferred with 1,000 ultrafast bootstrap (UFBoot) replicates to assess branch support. Numbers shown on the branches correspond to UFBoot support values. Molecular OTUs (mOTUs) identified in this study are labeled with the prefix "mOTU". Ultrafast bootstrap values are shown at the corresponding nodes. *Longiorus elongatus* was used as the outgroup.

*trichiura* (n = 22), followed by *N. americanus* + *T. trichiura* + *Ascaris* (n = 12). Remarkably, one individual was infected with all four detected nematode species. Notably, no single infections with *S. stercoralis* were observed; this species was detected only in individuals co-infected with other geohelminths (Fig 5).

### *Entamoeba* and other amoebae

In the NGS metataxonomic analysis, species-level assignment was performed using a phylogenetic approach. This allowed us to confirm the presence of *E. coli*, *E. hartmanni*, and *E. dispar* in the studied population, with strong phylogenetic support (UFB ≥ 98) for each clade (S2 Fig).

A total of 34 individuals (52.3%) tested positive for any *Entamoeba* species by microscopy, while 42 individuals (64.6%) were positive using NGS-based metataxonomics. Overall, NGS outperformed microscopy in detecting *Entamoeba* species. The most striking difference was observed for *E. hartmanni*, with only 9 positives identified by microscopy compared to 29 detected by NGS—over three times more. For *E. coli*, the results were more similar, with NGS detecting 25 positives and microscopy 24. In the case of *E. histolytica/dispar/moshkovskii*, both methods yielded identical results, with 13 positive individuals each (Fig 6). Interestingly, the phylogenetic analyses resolved the amoeba diversity at the species level,

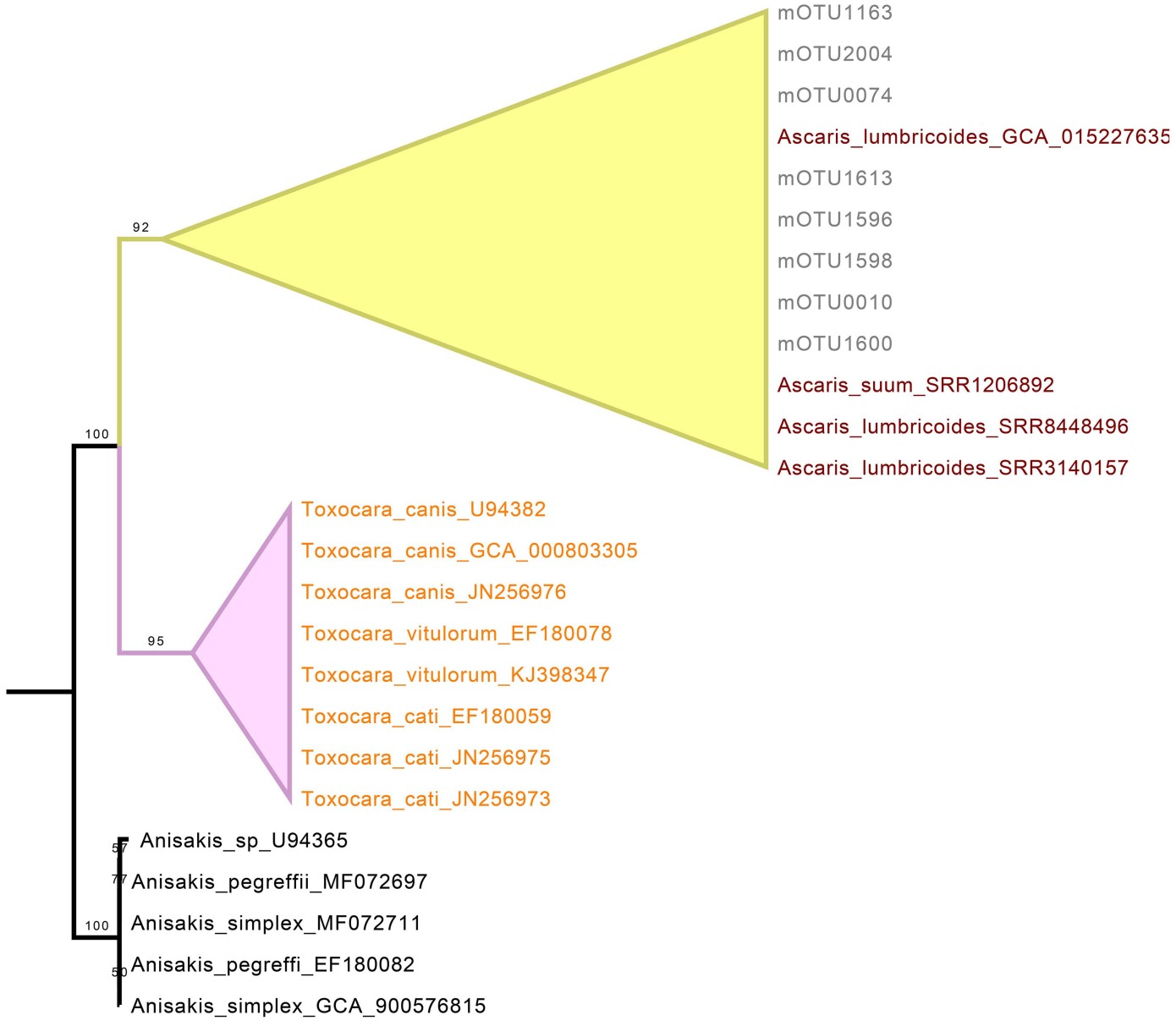

**Fig 3. Phylogenetic analysis of Ascaris mOTUs for taxonomic assignment.** Maximum-likelihood phylogenetic tree based on the SSU rRNA gene, constructed using a curated selection of nematode sequences from the genera *Ascaris*, *Toxocara* and *Anisakis*. Numbers shown on the branches correspond to UFBoot support values. The tree was inferred with 1,000 ultrafast bootstrap (UFBoot) replicates to assess branch support. Molecular OTUs (mOTUs) identified in this study are labeled with the prefix "mOTU". Ultrafast bootstrap support values are shown at the respective nodes. *Anisakis* genus was used as the outgroup.

consistent with the current species framework for this genus, with bootstrap support values that segregated closely related taxa, including members of the *Entamoeba histolytica/dispar/moshkovskii* complex.

The microscopy examination identified four positive samples for *Iodamoeba*, whereas no detections were observed using the NGS-based approach. This discrepancy is likely due to poor primer binding, as the reverse primer used for PCR shows limited annealing efficiency in *Iodamoeba* due to low sequence conservation in the targeted rDNA region.

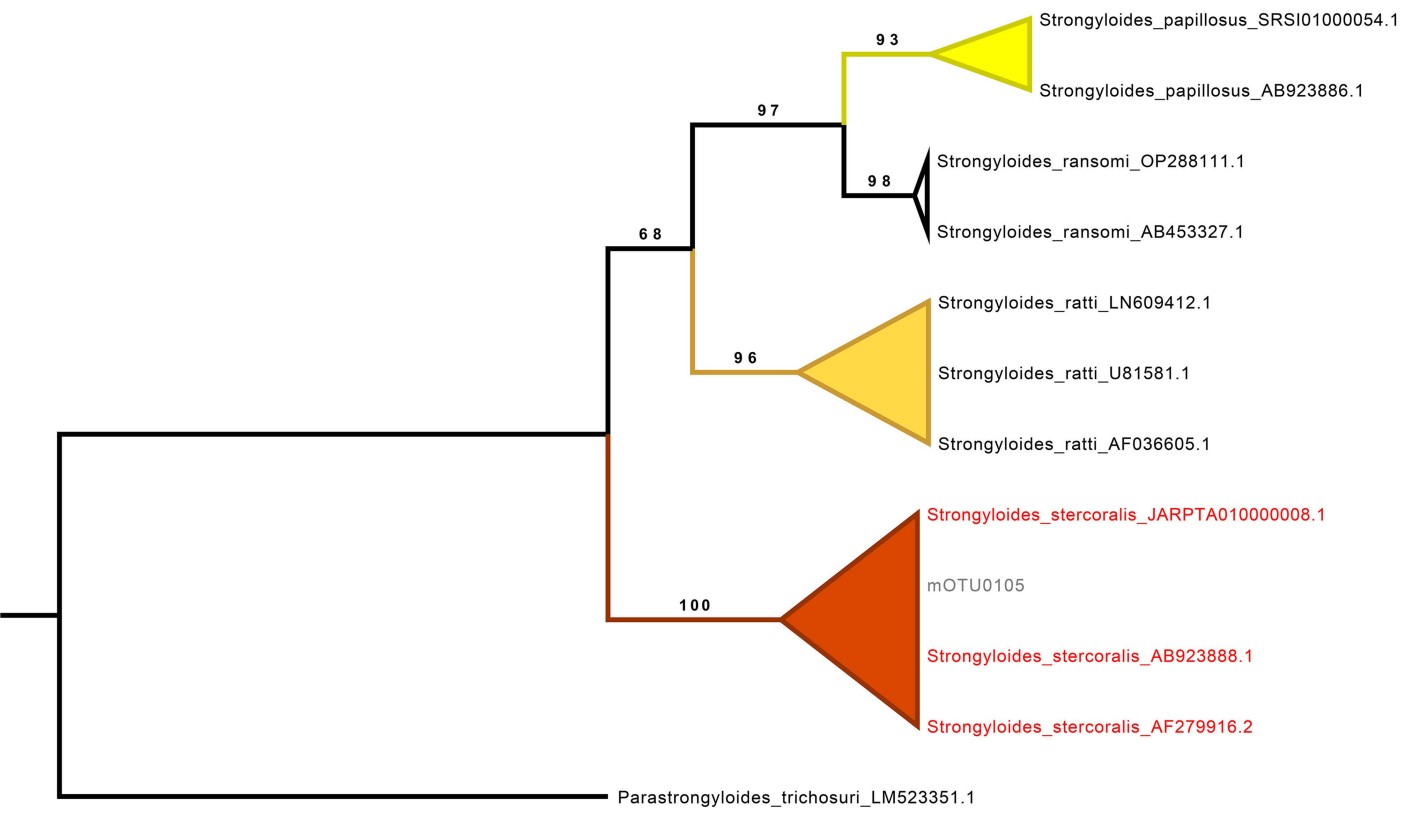

**Fig 4. Phylogenetic analysis of *Strongyloides* mOTUs for taxonomic assignment.** Maximum-likelihood phylogenetic tree based on the SSU rRNA gene, constructed using a curated selection of *Strongyloides* species sequences. The tree was inferred with 1,000 ultrafast bootstrap (UFBoot) replicates to assess branch support. Numbers shown on the branches correspond to UFBoot support values. Molecular OTUs (mOTUs) identified in this study are labeled with the prefix "mOTU". Ultrafast bootstrap support values are shown at the corresponding nodes. *Parastrongyloides trichosuri* was used as the outgroup.

Based on the higher specificity of the NGS results, we examined the distribution of *Entamoeba* species and their co-occurrence patterns within the studied population. While single-species infections involving any of the three detected *Entamoeba* species (*E. coli*, *E. hartmanni*, and *E. dispar*) were observed, mixed infections involving all possible species combinations were also detected. The most common infection types were single *E. hartmanni* infections and mixed *E. coli* + *E. hartmanni* infections, with 12 cases each (Fig 7).

## Blastocystis

*Blastocystis* represents another compelling case in which microscopy significantly underestimated the number of colonized individuals. While Parasep concentration and direct stool smear analyses detected only 4 positive samples, the NGS metataxonomic approach identified *Blastocystis* in 46 individuals (70.7%), an eleven-fold increase (Fig 8).

Subtype classification was performed as described in the Methods section, using the mOTUs assigned to *Blastocystis*. As shown in the phylogenetic tree, all detected mOTUs were assigned to subtypes ST1, ST2, and ST3, with robust phylogenetic support (UFB 100) (S3 Fig).

The most frequent colonization pattern involved a single subtype: ST1 (n = 13), followed by ST3 (n = 10) and ST2 (n = 9). Mixed subtype colonization was observed in 14 individuals (21.5%), with ST1 + ST3 being the most common combination (n = 8). Notably, triple subtype colonization (ST1 + ST2 + ST3) was detected in 2 individuals (Fig 9).

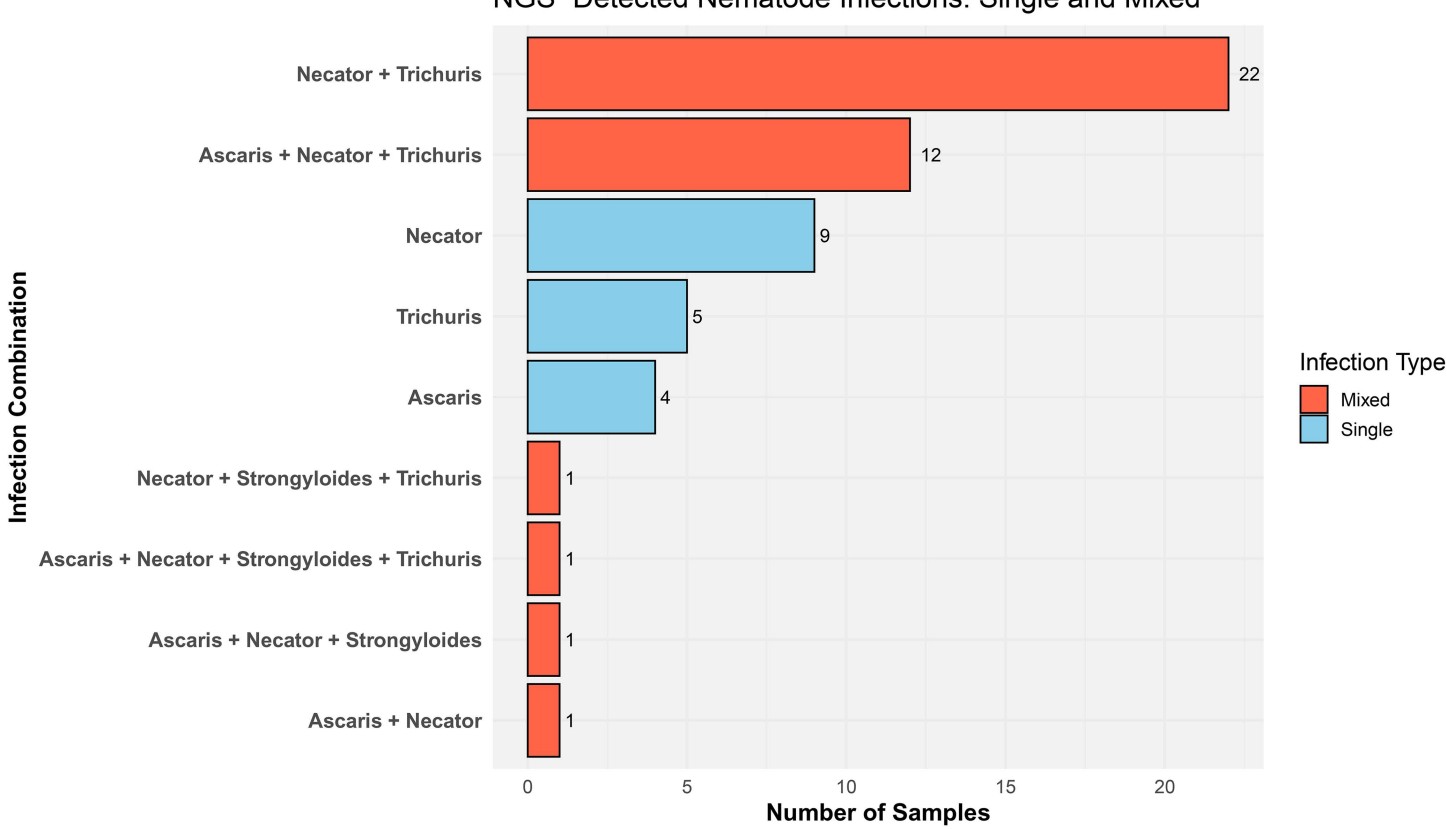

**Fig 5. Distribution of single and mixed nematode infections detected by Next-Generation Sequencing (NGS).** Horizontal bar plot summarizing infection combinations detected by NGS across all samples for four soil-transmitted helminths: *Ascaris spp.*, *Necator americanus*, *Strongyloides stercoralis*, and *Trichuris trichuria*. Each bar represents the number of samples exhibiting a specific infection pattern (X-axis), categorized as either single-species (Single) or mixed-species infections (Mixed). Bars are color-coded by infection type: blue for single infections and red for mixed infections.

## Other protozoa

Regarding the other protozoan intestinal parasites, the NGS metataxonomic approach detected only *Cystoisospora*. Four mOTUs were assigned to this coccidian genus. Phylogenetic analysis showed that these mOTUs formed a monophyletic group with known mammalian parasitic species including *Cystoisospora belli*, *Cystoisospora ohionensis*, *Cystoisospora suis*, *Cystoisospora canis*, and *Cystoisospora felis*, with bootstrap support (98%) (Fig 10). This phylogenetic placement supports the genus-level identity. Microscopy reported the presence of *Chilomastix*, *Cyclospora*, and *Giardia*, while NGS metataxonomics detected only *Cystoisospora*, which was found in two individuals. Noteworthily, the two *Cystoisospora*-positive individuals were negative for *Cyclospora*, the other coccidian parasite (Fig 11).

## Discussion

Intestinal parasites remain a significant public health burden in low- and middle-income countries and are increasingly recognized as an emerging concern in developed nations [50–52]. Since the initial discovery of intestinal protists and nematodes, microscopy has served as the foundation for the diagnosis of intestinal parasitic infections in both humans and animals [53,54] and due to the challenges associated with culturing many intestinal parasites under controlled laboratory conditions [55], still persist in many laboratories as the primary method for taxonomic classification.

## Comparison of Amoebae Detection: NGS vs Microscopy

**Fig 6. Comparison of amoebae detection using NGS-Metataxonomics and microscopy.** Bar plot showing the number of positive samples for intestinal amoebae: *Entamoeba dispar*, *Entamoeba coli*, *Entamoeba hartmanni*, and *Iodamoeba* sp., detected using either NGS-Metataxonomics (NGS) or microscopy (Mic). Bars represent the total number of detections for each parasite by method across all samples (Y-axis).

Intestinal protozoa and nematode infections occur through interactions between humans, their domestic animals—both companion and livestock— and the surrounding environment. Molecular studies already show parasite exchange across species barriers. Comparative analyses of *Strongyloides* 18S and cox1 sequences from humans and dogs have revealed two major *S. stercoralis* lineage, one apparently restricted to canines, and another shared among dogs, cats, humans, and non-human primates [56]. These findings indicated a limited yet plausible zoonotic link, though the direction and geographic extent of transmission remain uncertain, underscoring the need for co-sampling of hosts and population genetic analyses to clarify cross-species dynamics. Human-associated *Blastocystis* subtypes (ST1-ST3) are frequently detected in cohabiting pets and their owners, suggesting either bidirectional transmission or exposure to shared environmental sources [57]. Given that most studies are cross-sectional, current evidence provides only limited insights into parasite transmission, highlighting the need for high-resolution genotyping, longitudinal sampling, and environmental assessments. Similarly, genetic analyses show extensive overlap among soil-transmitted helminths in humans and animals, suggesting shared transmission cycles rather than exclusively human reservoirs [8]. These findings indicate the need for integrated parasite control and management to mitigate zoonotic and environmental risks. Monitoring and preventive measures across veterinary, medical, and public health sectors are critical to reduce cross-species transmission, but current evidence reveals gaps in treatment, evaluation and coordinated efforts involving health and veterinary professionals, researchers, industry, and governmental authorities [58]. In this context, metataxonomic approaches offer a powerful tool to advance One Health investigations, by providing high-resolution data on parasite diversity, geographic distribution, and potential reservoirs, supporting informed, evidence-based interventions across humans, animals, and the environment.

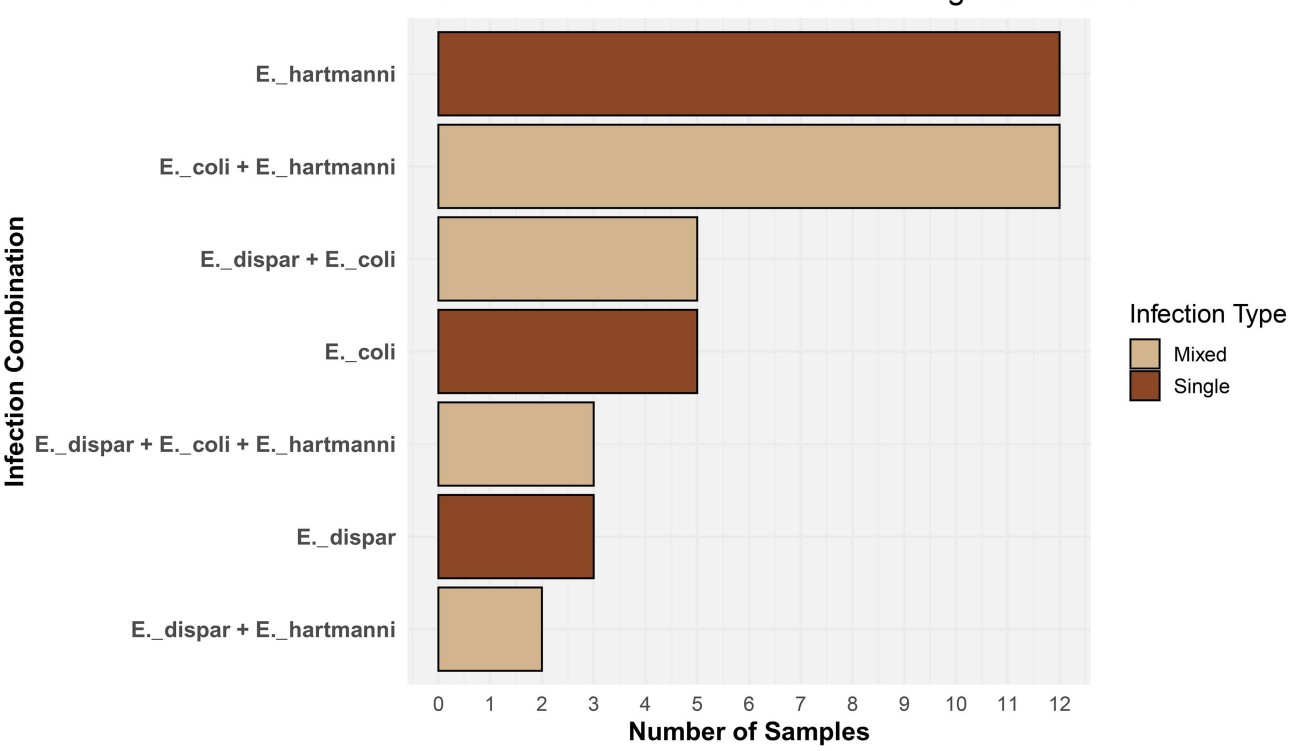

**Fig 7. Distribution of single and mixed amoeba infections detected by NGS-Metataxonomics.** Horizontal bar plot summarizing infection combinations detected by NGS across all samples for intestinal amoebae: *Entamoeba dispar*, *Ent. coli*, *Ent. hartmanni*. Each bar represents the number of samples exhibiting a specific infection pattern (X-axis), categorized as either single-species (Single) or mixed-species (Mixed) infections. Bars are color-coded by infection type: brown for single infections and tan for mixed infections. Parasite names were abbreviated for clarity (e.g., Entamoeba→E.).

The advent of PCR in the 1980s and 1990s marked a major advancement, improving the sensitivity and specificity of diagnostic tests and enabling more refined classification of parasitic organisms [10,11,16]. However, recent biotechnological innovations have continued to drive progress across all areas of life sciences. Notably, next-generation sequencing (NGS) technologies have revolutionized parasitology by enabling high-throughput detection methods such as metataxonomic analysis [25,26,34,39]. Unlike traditional qPCR-based methods that rely on fluorescence detection of specific amplicons, NGS allows for the sequencing of hundreds of thousands of DNA fragments simultaneously [39]. This dramatically enhances detection sensitivity and, when coupled with modern bioinformatic tools and phylogenetic analyses, significantly improves taxonomic resolution and specificity [41].

In this study, we combined a metataxonomic approach with classical phylogenetic methods to detect and classify intestinal parasites in rural populations of Colombia, where high prevalence rates have been historically documented [59]. While metataxonomic methods have been increasingly applied for the detection of protist parasites [24–26,34], our study expanded their use to include the detection of geohelminths, including *S. stercoralis*, and, in certain cases, suggested potential species-level resolution within a curated phylogenetic framework. Additionally, we aimed to compare the performance of traditional microscopy with that of NGS-based metataxonomics, highlighting the strengths and limitations of each method in accurately characterizing parasitic infections. Although both technologies successfully detected intestinal nematodes, notable differences were observed in certain cases. For *Trichuris trichiura*, microscopy identified significantly more infected individuals than the NGS-based metataxonomic approach. A plausible explanation is the high resistance of

**Fig 8. Comparison of *Blastocystis* detection using NGS-Metataxonomics and Microscopy.** Bar plot showing the number of samples positive for *Blastocystis hominis* detected by either NGS-Metataxonomics (NGS) and by conventional microscopy (Mic). Each bar represents the total number of positive samples identified by each diagnostic method (Y-axis).

the *Trichuris* eggshell [60], which may hinder efficient DNA extraction and reduce the yield of amplifiable material. A similar, albeit less pronounced, pattern was observed for *Ascaris*, with microscopy detecting one additional positive case. We applied a physical and chemical method for lysis according to manufacturer instructions. Future studies should implement improved lysis protocols to enhance the disruption of tough nematode eggshells [61] and increase detection sensitivity for these species. An additional consideration with *Ascaris* is that since *A. lumbricoides* and *A. suum* are closely related species [62], the 18S rRNA gene does not provide sufficient resolution to distinguish between them, limiting species-level assignment in metataxonomic analyses.

In contrast, for hookworms and *S. stercoralis*—both of which possess less resistant external structures [63]—microscopy did not outperform metataxonomics. In fact, NGS-based metataxonomic analysis proved more effective for *S. stercoralis*, detecting three times as many positive samples. This is particularly relevant given the clinical importance of *S. stercoralis*, which can cause severe, life-threatening infections, especially in immunocompromised individuals [52]. Another advantage of using NGS technologies is their ability to reduce the risk of misidentifying *Strongyloides* rhabditoid larvae as hookworm larvae. Traditional parasitological methods for its detection (e.g., direct smear, Baermann, agar plate culture) are labor-intensive, technically demanding, and have limited sensitivity—often detecting fewer than 60–70% of cases in single-sample analyses [64]. Molecular approaches, including NGS, offer significantly improved sensitivity and can effectively overcome the challenges posed by low and intermittent larval shedding [64].

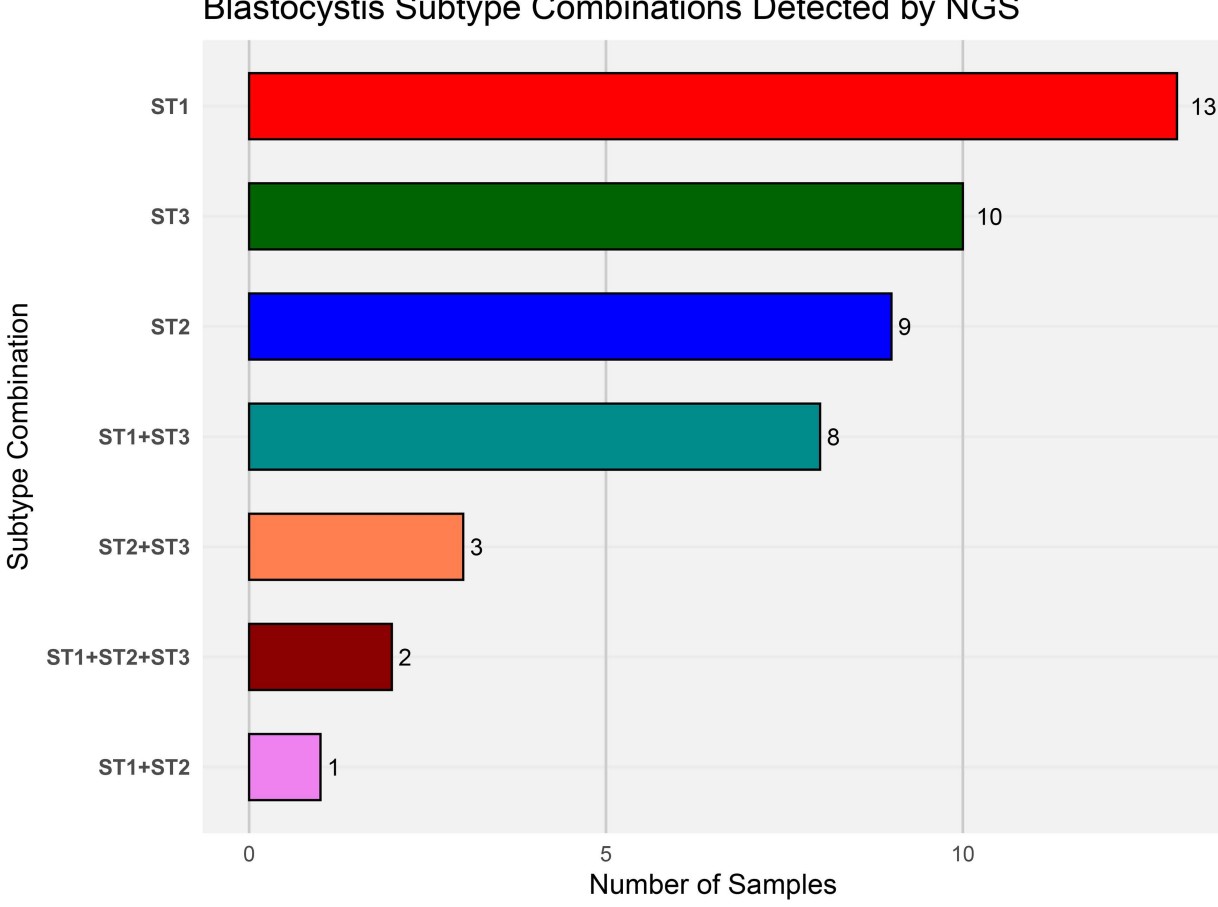

**Fig 9. Distribution of *Blastocystis* subtype (ST) combinations detected by NGS-Metataxonomics.** The bars represent the occurrence of single and mixed Blastocystis subtypes among positive samples (X-axis).

One potential advantage of the metataxonomic approach lies in its capacity, when integrated with phylogenetic analyses, to provide putative species-level assignments for several intestinal parasites. However, such classifications should be interpreted with caution, as they are limited by the phylogenetic resolution of the ribosomal marker used and by the completeness of current taxonomic reference databases. As NGS technologies continue to advance and genomic databases for parasites become more comprehensive, the inclusion of multiple molecular markers—or even full phylogenomic approaches—will enable more confident species-level identifications within a robust and resolved taxonomic framework. This represents a substantial improvement over microscopy, which typically permits only genus-level identification and therefore restricts our understanding of the true epidemiology, burden, and transmission dynamics of parasitic infections. By applying our combined NGS-based metataxonomic and phylogenetic framework, we found evidence that the Colombian populations analyzed were infected, in descending order of frequency, with *T. trichiura*, *N. americanus*, *Ascaris* spp., and *S. stercoralis*.

In the case of protist detection, the NGS-based metataxonomic approach has been successfully applied in several studies involving human stool samples and wastewater. Among the protists reliably detected using this technology are *Blastocystis*, *Giardia*, *Entamoeba*, *Balantioides*, *Acanthamoeba*, *Cryptosporidium*, *Dientamoeba*, and *Rhogostoma* [24,26,65]. These examples highlight the broad taxonomic scope and sensitivity of metataxonomic methods for detecting

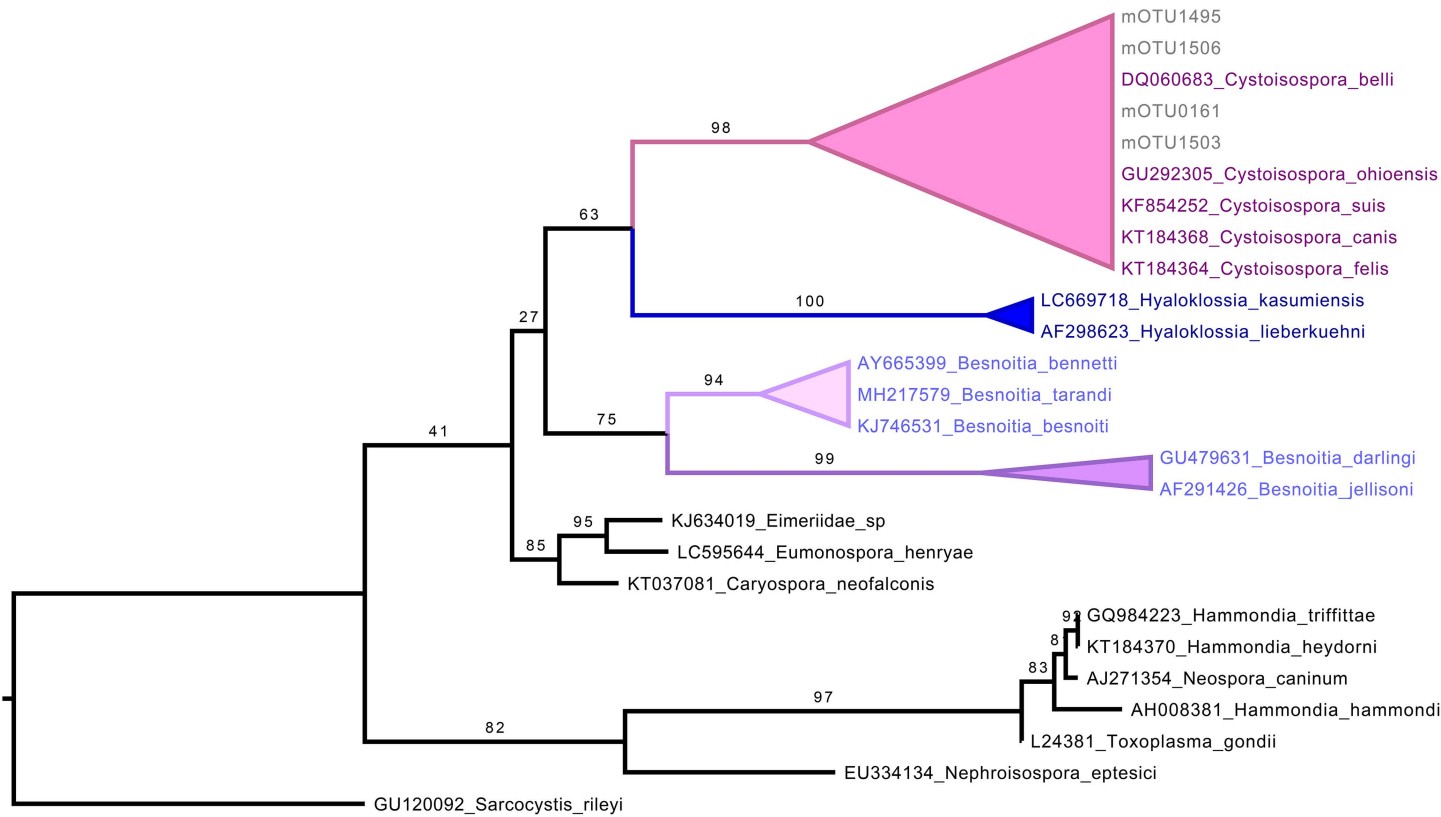

**Fig 10. Phylogenetic analysis of *Cystoisospora* mOTUs for taxonomic assignment.** Maximum-likelihood phylogenetic tree based on the 18S rRNA gene, constructed using a curated selection of reference sequences from apicomplexan parasites, including representatives of the genus *Cystoisospora*. The tree was inferred with 1,000 ultrafast bootstrap (UFBoot) replicates to assess branch support. Numbers shown on the branches correspond to UFBoot support values. Molecular OTUs (mOTUs) identified in this study are labeled with the prefix "mOTU". Ultrafast bootstrap support values are shown at the corresponding nodes. *Sarcocystis rileyi* was used as the outgroup.

a wide range of eukaryotic microorganisms in complex biological and environmental samples. As shown in previous studies worldwide [24,66,67], *Blastocystis* is the most prevalent intestinal protist in humans. Our study reflected the same trend, with *Blastocystis* detected in nearly 71% of the tested population using NGS-based metataxonomic technologies. Strikingly, the sensitivity of microscopic methods was notably lower, as metataxonomics detected over eleven times more colonized individuals than traditional smear techniques.

A further potential advantage of the metataxonomic plus phylogenetic approach is its capacity to assign *Blastocystis* sequences to specific subtypes and to identify possible mixed-subtype infections [24]. Our findings align with previous studies by our group and others, indicating that the dominant subtypes colonizing Colombian populations are ST1, ST2, and ST3, with ST1 being the most prevalent. Additionally, we found that mixed-subtype colonization was common, occurring in approximately 22% of positive individuals. The most frequent combination was ST1 + ST3, and two individuals harbored all three subtypes simultaneously.

The second most prevalent genus detected was *Entamoeba*, with nearly 65% of individuals testing positive by NGS-based metataxonomics. In contrast, microscopic smear analysis detected *Entamoeba* in 52% of the individuals. The most striking discrepancy between the two methods was observed at the species level: in our study, *E. hartmanni* appeared to be underdiagnosed by microscopy, with NGS detecting three times more cases. This underestimation may be due to morphological misidentification or confusion with other protists under light microscopy [68]. For *Entamoeba coli* and *E.*

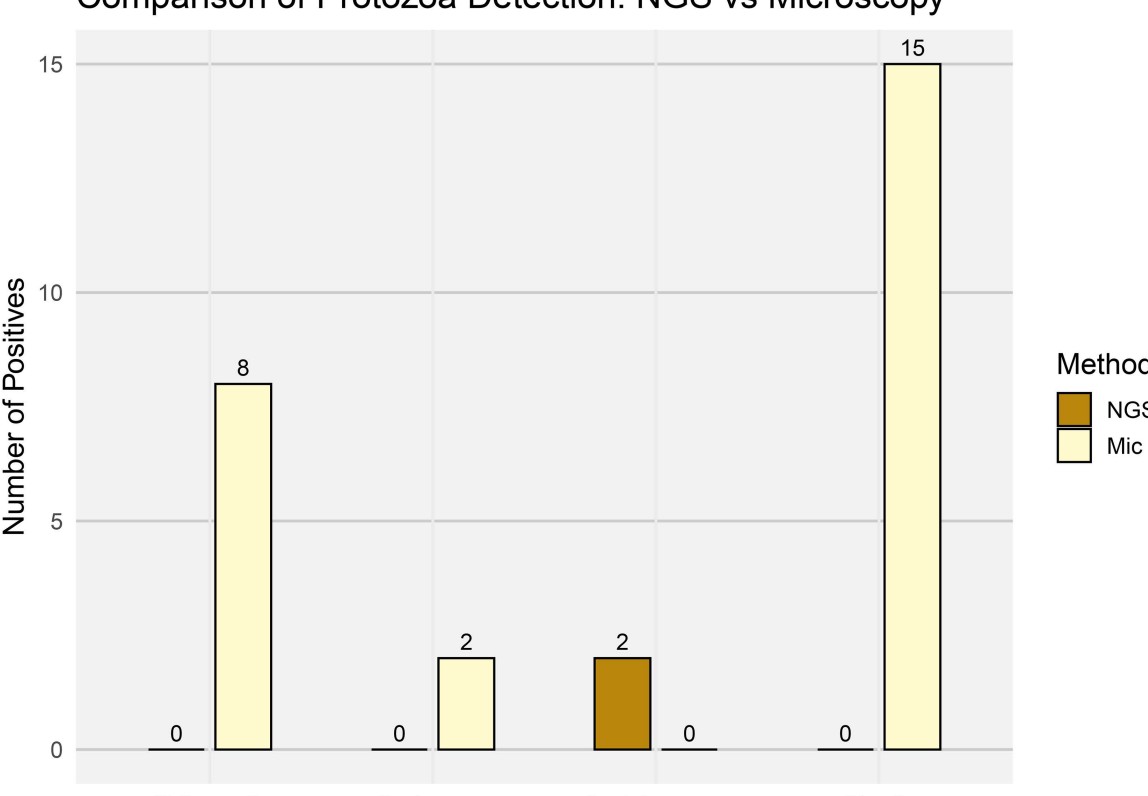

**Fig 11. Comparison of other protozoan parasite detection by NGS-Metataxonomics and microscopic smear analysis.** Bar plot summarizing the number of positive samples for other intestinal protozoa (*Giardia*, *Cyclospora*, *Cystoisospora*, and *Chilomastix*) as detected by NGS-Metataxonomics (NGS) and microscopy (Mic). Each bar represents the total number of samples testing positive per parasite and method (Y-axis).

*dispar*, both methods showed similar performance. In microscopy, the amoeba cysts identified as part of the *Entamoeba histolytica/dispar/moshkovskii* complex were classified as *Entamoeba dispar* in the NGS analysis, thanks to the resolution provided by phylogenetic inference. The similar number of positive samples obtained by both methods suggests comparable performance in detecting this genus.

An additional discordance was noted with *Iodamoeba*: microscopy reported four positive cases, whereas NGS-based metataxonomics failed to detect any. We suspect this is due to inefficient amplification caused by mismatches between the reverse primer and the 18S-V4 target region in *Iodamoeba*, leading to poor PCR performance during library preparation. Overall, the population studied showed colonization primarily by non-pathogenic, commensal amoebae. Phylogenetic analysis revealed strong resolution within the *E. histolytica/E. dispar/E. moshkovskii* species complex, and we can rule out the presence of the pathogenic *E. histolytica* in the detected *Entamoeba* mOTUs. Our results agree with PCR-based molecular studies in Colombian population which have reported a higher circulation of *E. dispar* compared with *E. histolytica* [69].

The most discordant results were observed for other protozoa, namely *Chilomastix*, *Cyclospora*, *Cystoisospora*, and *Giardia*. For these protozoan taxa, there was no concordance between microscopy and NGS metataxonomic; each was detected by only one method, with no mutual confirmation. This highlights the inherent challenge of capturing the full diversity of parasites using a single methodological approach, given the varying biological and molecular characteristics of these organisms. In 18S rRNA metataxonomics, several technical factors may contribute to false-negative results. First,

DNA extraction protocols may not fully disrupt the resistant cyst or oocyst walls of certain protozoa, limiting the recovery of amplifiable DNA [70]. Second, sequence polymorphisms in the target regions can affect the binding efficiency of universal primers, introducing amplification bias [71]. Third, the low abundance of parasite DNA relative to host and microbiome DNA can reduce amplification efficiency [72]. Additionally, variability in ribosomal gene copy number among taxa and preferential amplification of shorter fragments, particularly when targeting V3-V4 regions, can further affect detection sensitivity [73,74]. Taken together, these factors indicate that absence of detection by NGS should be interpreted cautiously, as it may reflect technical limitations rather than true absence of the organism.

In the case of *Chilomastix*, bioinformatic analysis revealed an issue similar to that observed with *Iodamoeba*: the reverse primer target region shows poor sequence conservation in this genus, likely impairing efficient primer annealing and thus detection by PCR. For *Cyclospora*, although we cannot entirely rule out primer incompatibility, the same primer set has successfully amplified other apicomplexan parasites (e.g., *Cystoisospora*, *Cryptosporidium*) in past analyses. Therefore, no definitive conclusion can be made regarding *Cyclospora* detection.

Interestingly, *Giardia* was not detected by NGS in any of the samples analyzed in this study, even though we have previously reported its successful detection using the same primers and protocols in human stool samples and wastewater from Colombia [41]. This observation is consistent with other metataxonomic studies that have reported limited or no detection of *Giardia*, highlighting persistent technical challenges [22,34,75,76]. Several authors have suggested that primer mismatches, low abundance, and especially inefficient DNA extraction from the parasite's resistant cyst wall may underline these failures. Standard fecal DNA extraction protocols may be suboptimal for *Giardia* which often require more intensive lysis methods— such as mechanical disruption or repeated freeze-thaw cycles— to ensure adequate DNA release [77].

Other molecular approaches, such as PCR, have been employed to detect intestinal parasites. However, these techniques present inherent limitations, especially when aiming to identify a wide range of taxa. Detecting more than 20 different parasite genera—each comprising multiple species or subtypes, as in the case of *Blastocystis*—requires the use of multiple primer sets and separate reactions. This not only reduces scalability but also increases cost and complexity over time. In contrast, the NGS-based metataxonomic approach enables the simultaneous detection of a broad and potentially unlimited diversity of intestinal parasites in a single assay. Furthermore, the downstream analysis can be streamlined and automated using current, well-established bioinformatic pipelines.

Next-generation sequencing (NGS) has not replaced conventional diagnostic tools but has experienced growing adoption as a complementary methodology in parasitology. Its applications have expanded significantly beyond detection to include understanding genetic interrelationships among parasites, characterizing genetic diversity and intra-isolate variation, and determining the relative abundance of specific parasite species within complex clinical samples [74]. The available research demonstrates several key comparative advantages of NGS over traditional diagnostic tools for parasite detection, though comprehensive head-to-head studies remain limited. When comparing next-generation sequencing (NGS) to PCR in the diagnosis of parasitic infections, NGS offers broader detection capabilities, particularly for mixed infections and subtype resolution, while PCR tends to provide higher sensitivity for single-target detection. For example, in studies on *Blastocystis* spp., qPCR was more sensitive for detecting the presence of the parasite in low-load samples, but NGS was superior in identifying multiple subtypes and mixed infections, which were undetected by PCR and Sanger sequencing [78]. Stensvold et al. directly compared metabarcoding (NGS) and real-time PCR (qPCR) for detecting zoonotic protists in pigs, revealing complementary methodological strengths [76]. While both methods were equally effective for detecting *Balantioides coli*, metabarcoding proved superior for broad-spectrum screening, simultaneously identifying additional parasites like *G. intestinalis* and *Enterocytozoon bieneusi*. In contrast, qPCR remained the optimal tool for sensitive and targeted quantification of specific pathogens. In relation to intestinal amoebae, several studies have demonstrated that next-generation sequencing (NGS) offers markedly improved sensitivity and resolution over conventional methods such as microscopy and PCR. NGS-based approaches, including amplicon sequencing and metagenomics,

consistently detect higher species richness, mixed infections, and intra-host diversity that are frequently overlooked by traditional diagnostics [79,80]. For instance, short-amplicon NGS revealed substantial genetic diversity within *Iodamoeba bütschlii*, including host-specific lineages and novel variants undetectable by standard PCR [79]. Similarly, NGS analyses of *E. coli*, *E. hartmanni*, and *Endolimax nana* have uncovered extensive intrageneric diversity and previously unrecognized ribosomal lineages [80,81]. In livestock and wildlife hosts, NGS has identified mixed-subtype infections and novel genotypes of *Entamoeba* and *Blastocystis*, expanding the known host range and genetic landscape of these protists [34,80].

NGS has proven particularly valuable for protozoan pathogens such as *G. intestinalis* and *Cryptosporidium* spp., where mixed infections are common but often undetected by conventional sequencing methods. For *Giardia*, several studies have demonstrated that next-generation amplicon sequencing (NGS) provides significant methodological advantages over conventional Sanger sequencing for molecular characterization and epidemiological investigations [33,82,83]. NGS consistently shows higher sensitivity and improved capacity to detect mixed assemblage infections, which are frequently missed or yield ambiguous results with Sanger sequencing due to its inability to resolve heterogeneous sequences. By targeting loci such as the beta-giardin and tpi genes, deep amplicon sequencing has revealed the presence and relative abundance of multiple assemblages (e.g., A, B, AII, BIV) within individual hosts and outbreak samples [33,82,83]. Notably, in outbreak investigations, NGS successfully identified mixed assemblages in 43.5% of cases where Sanger failed and further detected 12 novel sub-assemblage-level variants [83]. Additionally, its ability to quantify minor single-nucleotide variants present at frequencies as low as 5% provides unprecedented resolution into parasite population structure. These capabilities not only offer deeper genetic insights but also enhance the precision of transmission tracking and public health surveillance efforts. Furthermore, in the case of *Cryptosporidium*, NGS offers a critical advantage by revealing a complex landscape of infection that remains hidden to conventional methods. While Sanger sequencing of the gp60 gene often identifies only a dominant subtype, NGS applications consistently uncover mixed infections and substantial within-subtype genetic diversity [29,84,85].This high-resolution view of the pathogen population is crucial for accurate epidemiological tracing and for understanding the true dynamics of transmission.

Our current metataxonomic approach fails to detect cestodes and trematodes, likely due to poor primer annealing caused by the high sequence divergence of their 18S rDNA V4 region compared to other intestinal parasites such as protists and nematodes. Bioinformatic analyses from this study indicate that Platyhelminths are not efficiently amplifiable with the primers used. Additionally, certain amoebae—such as *Iodamoeba*—appear to be inefficiently amplified, likely due to mismatches at the primer binding sites. These findings highlight the need for improved primer design to broaden taxonomic coverage and enhance the diagnostic utility of metataxonomic protocols.

Another potential limitation of this study is the long-term storage of DNA samples prior to NGS analysis. While prolonged preservation at -80°C generally maintains DNA integrity, factors such as storage duration and freeze-thaw cycles can influence quality. Evidence from microbiota research indicates that fecal DNA can remain viable for extended periods, even up to 14 years, under stable frozen conditions [86–88]. However, parasite specific data is more limited. Although studies suggest that parasite DNA, such as from *Giardia*, remains detectable after long-term storage [89], the effect on the broader parasitome is less clear. In our study, all samples were maintained under a strict and uniform storage protocol at -80°C from collection until DNA extraction. This consistent handling ensures that any potential DNA degradation would be systematic across the entire sample set. Consequently, such effects would likely reduce overall detection sensitivity uniformly, without compromising our core finding that NGS identifies greater parasite diversity than microscopy in these archival samples.

We are now entering a new era in parasitology, empowered by an expanded molecular toolkit that includes genomic, metagenomic, and metataxonomic approaches. These technologies provide high-resolution taxonomic and evolutionary insights, enable the genomic characterization of unculturable parasites, and enhance the detection of parasitic organisms in complex samples such as stool or environmental matrices [90].

## Supporting information

**S1 Fig. Phylogenetic analysis of *Necator* mOTUs for taxonomic assignment.** Maximum-likelihood phylogenetic tree based on the 18S rRNA gene, constructed using a curated selection of reference sequences from nematode genera related to hookworms, including representatives of *Necator* and *Ancylostoma*. The tree was inferred with 1,000 ultrafast bootstrap (UFBoot) replicates to assess branch support. Molecular OTUs (mOTUs) identified in this study are labeled with the prefix "mOTU". All hookworm mOTUs clustered within the *Necator americanus* clade, indicating that this was the sole hookworm species detected in the sampled population. Ultrafast bootstrap support values are shown at the corresponding nodes. Rhabditis sp. was used as the outgroup.
(PDF)

**S2 Fig. Phylogenetic analysis of *Entamoeba* mOTUs for taxonomic assignment.** Maximum-likelihood phylogenetic tree based on the 18S rRNA gene, constructed using a curated selection of reference sequences from the genus *Entamoeba*. The tree was inferred with 1,000 ultrafast bootstrap (UFBoot) replicates to assess branch support. Molecular OTUs (mOTUs) identified in this study are labeled with the prefix "mOTU". All mOTUs clustered within known *Entamoeba* species clades, supporting their taxonomic assignments. Ultrafast bootstrap support values are indicated at the corresponding nodes. Representative sequences of *Dientamoeba fragilis* were used as the outgroup.
(PDF)

**S3 Fig. Phylogenetic analysis of *Blastocystis* mOTUs for subtype (ST) classification.** Maximum-likelihood phylogenetic tree based on the 18S rRNA gene, constructed using a curated selection of *Blastocystis* reference sequences representing subtypes (STs) 1–17, including all subtypes commonly associated with human populations. Molecular OTUs (mOTUs) identified in this study are labeled with the prefix "mOTU". The tree was inferred with 1,000 ultrafast bootstrap (UFBoot) replicates to assess branch support, and UFBoot values are shown at the corresponding nodes. All mOTUs clustered within well-supported clades corresponding to known *Blastocystis* subtypes, enabling confident subtype assignment. A combined clade of *Blastocystis* ST15 and ST17 was used as the outgroup.
(PDF)

**S1 Table. Description of the conventional parasitological results.**
(XLSX)

**S2 Table. Read statistics for the bacterial 16S rRNA amplicon sequencing and processing.**
(XLSX)

**S3 Table. Accession numbers for the reference sequences.**
(XLSX)

## Author contributions

**Conceptualization:** Ana L. Galvan-Diaz, Gisela M. Garcia-Montoya, Juan F. Alzate.

**Data curation:** Katherine Bedoya-Urrego, Nicolas Rozo-Montoya, Gisela M. Garcia-Montoya, Juan F. Alzate.

**Formal analysis:** Katherine Bedoya-Urrego, Nicolas Rozo-Montoya, Ana L. Galvan-Diaz, Gisela M. Garcia-Montoya, Juan F. Alzate.

**Funding acquisition:** Ana L. Galvan-Diaz, Gisela M. Garcia-Montoya, Juan F. Alzate.

**Investigation:** Katherine Bedoya-Urrego, Nicolas Rozo-Montoya, Ana L. Galvan-Diaz, Gisela M. Garcia-Montoya, Juan F. Alzate.

**Methodology:** Katherine Bedoya-Urrego, Nicolas Rozo-Montoya, Ana L. Galvan-Diaz, Gisela M. Garcia-Montoya, Juan F. Alzate.

**Project administration:** Juan F. Alzate.

**Resources:** Juan F. Alzate.

**Software:** Nicolas Rozo-Montoya, Juan F. Alzate.

**Supervision:** Juan F. Alzate.

**Validation:** Katherine Bedoya-Urrego, Ana L. Galvan-Diaz, Juan F. Alzate.

**Visualization:** Juan F. Alzate.

**Writing – original draft:** Katherine Bedoya-Urrego, Ana L. Galvan-Diaz, Juan F. Alzate.

**Writing – review & editing:** Katherine Bedoya-Urrego, Nicolas Rozo-Montoya, Ana L. Galvan-Diaz, Gisela M. Garcia-Montoya, Juan F. Alzate.

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
