## [Decision Letter · Decision Letter 0]

24 Sep 2025

Dear Dr. Alzate,

Thank you for submitting your manuscript to PLOS ONE. After careful consideration, we feel that it has merit but does not fully meet PLOS ONE’s publication criteria as it currently stands. Therefore, we invite you to submit a revised version of the manuscript that addresses the points raised during the review process.

We look forward to receiving your revised manuscript.

Kind regards,

Petr Heneberg

Academic Editor

PLOS ONE

Journal Requirements:

“This study was funded by Escuela de Microbiología-CODI, Universidad de Antioquia, under grant code 2023-64370.”

3. Please note that your Data Availability Statement is currently missing [the repository name and/or the DOI/accession number of each dataset OR a direct link to access each database]. If your manuscript is accepted for publication, you will be asked to provide these details on a very short timeline. We therefore suggest that you provide this information now, though we will not hold up the peer review process if you are unable.

“This study was funded by Escuela de Microbiología-CODI, Universidad de Antioquia, under grant code 2023-64370.”

“This study was funded by Escuela de Microbiología-CODI, Universidad de Antioquia, under grant code 2023-64370”

5. We note that there is identifying data in the Supporting Information file <Supp_TABLE_1_NGS_Mic.xlsx>. Due to the inclusion of these potentially identifying data, we have removed this file from your file inventory. Prior to sharing human research participant data, authors should consult with an ethics committee to ensure data are shared in accordance with participant consent and all applicable local laws.

-Location data

Additional Editor Comments:

The claim that metataxonomics outperforms microscopy is presented too strongly. The results show that sequencing was superior for detecting Strongyloides and Blastocystis, yet microscopy was more effective for Trichuris and Ascaris. This contrast demonstrates that the performance of each method is parasite dependent. The interpretation should therefore highlight complementarity rather than positioning one method as superior to the other.

The treatment of discrepancies between methods is not sufficiently critical. For example, the lack of Giardia detections in sequencing is attributed to true absence in the population, yet microscopy found positive cases. Technical limitations such as primer mismatch or inefficient DNA extraction provide more plausible explanations. Similar issues appear for Iodamoeba and Cyclospora. The manuscript would benefit from a stronger discussion of methodological causes for these mismatches, ideally supported by targeted molecular validation.

The sample size of 65 is small and combines specimens from two time periods almost seven years apart. Pooling across such a wide temporal gap introduces the possibility of confounding through epidemiological changes or laboratory practice differences. The authors should address this explicitly and discuss how these potential biases were controlled or accounted for in the study design.

Figures showing detection frequencies are presented as simple bar charts without any measure of uncertainty. Although presence and absence data are used, statistical analysis or variability indicators would allow readers to assess the strength of the comparisons. In the phylogenetic analyses, bootstrap values are presented as definitive, yet even high values need careful interpretation. A more cautious discussion of these analyses is necessary.

The methods section relies on a single primer pair targeting the 18S V4 region, which has known limitations. The exclusion of cestodes, trematodes, and some amoebae is acknowledged briefly, but the manuscript still describes the method as broad spectrum. A more nuanced description of primer biases and their impact on detection outcomes would improve the study’s credibility.

The raw sequencing data are stated to be deposited in the NCBI SRA under BioProject accession PRJNA1293464, with a reviewer link provided. This is an important positive step, and the final version should clearly include accession numbers to ensure compliance with data availability policies and to allow independent verification of results.

The discussion emphasizes advantages of sequencing approaches, yet does not sufficiently acknowledge the limitations of this marker or the need for complementary methods. For instance, species level identification of Ascaris is not possible with 18S rDNA, yet the text does not consistently highlight this limitation. Balancing strengths with weaknesses would provide a more rigorous and realistic assessment of the approach.

Ethical approval is reported and the use of archived anonymized samples is clearly justified. Consent and confidentiality procedures are explained adequately. However, the quality control procedures for microscopy are not described in detail. Including information on slide reading standards and technician validation would help rule out operator variability as a source of discrepancy.

The text is clearly structured and generally readable, yet it often repeats similar arguments and adopts a promotional tone. The writing would benefit from greater conciseness, less redundancy, and a more neutral presentation of the study’s impact. This would align the manuscript more closely with the standards of scientific reporting.

Reviewers' comments:

Reviewer's Responses to Questions

**Comments to the Author**

1. Is the manuscript technically sound, and do the data support the conclusions?

Reviewer #1: Yes

Reviewer #2: Partly

2. Has the statistical analysis been performed appropriately and rigorously?

Reviewer #1: N/A

Reviewer #2: Yes

3. Have the authors made all data underlying the findings in their manuscript fully available?

Reviewer #1: Yes

Reviewer #2: Yes

4. Is the manuscript presented in an intelligible fashion and written in standard English?

Reviewer #1: Yes

Reviewer #2: Yes

Reviewer #1: This is an interesting manuscript which describes the use of NGS for identifying parasites in clinical samples, focusing on Blastocystis and Entamoeba spp. The manuscript is well written and follows a logical structure, making it easy for readers to follow the flow of the study. The technical terminology used throughout is appropriate and no major grammatical errors were identified.

I felt that the introduction could be expanded to include more recent studies on NGS in parasitology but otherwise provided a good overview of the wider literature within the context of parasitology and molecular diagnostics.

The discussion was generally well structured but could benefit from the following minor revisions:

1) There is no mention of sequencing depth and coverage. Was this sufficient to detect low-abundance parasites? Insufficient sequencing depth could lead to under-detection of rare species or mixed infections. The authors should discuss how sequencing variability might influence the results.

2) The potential risk of contamination during sample processing or sequencing is not discussion. This is a common issue in NGS studies, potentially leading to false positives or biased results.

3) How can this be translated to clinical practice? Are the identified parasites clinically significant? What is the practicality of implementation in routine clinical/ field settings?

4) Discuss any recent studies on NGS applications in parasitology (if any). Include a more critical comparison of NGS with other emerging diagnostic technologies.

Reviewer #2: Intestinal parasites cause a major health burden worldwide, and traditional microscopy offers only limited sensitivity. In this manuscript, the authors applied an NGS-based metataxonomic approach, which showed higher sensitivity than microscopy for Strongyloides stercoralis but lower for Trichuris trichiura. The method also provided improved species- and subtype-level resolution. Overall, metataxonomics appears powerful for broad parasite detection, although primer design still requires refinement. I find this topic very interesting, but I have a few questions and suggestions to consider:

Major comments:

1. It is unclear whether random primers were used for library construction and sequencing, or whether DNA extraction was followed by library preparation employing only the 18S-V4Fw and 18S-V4 primers. An alternative possibility is that these primers were used specifically as supplements for sequencing. Clarification of this methodological step would be very helpful.

2. NGS is undoubtedly a powerful tool for species identification, but it also has the issue of noisy sequencing reads. The main concern here is that the ground truth of this study is not clearly defined, making it difficult to evaluate what are false positives or false negatives (e.g., lines 245–248) and how these were ruled out. In turns, it becomes challenging to assess which method truly outperforms the others.

3. The authors suggest that the failure of the NGS-based metataxonomic approach for Trichuris trichiura detection may be due to the high resistance of the eggshell, which leads to the issue of DNA isolation and lower yield. This hypothesis could be easily tested by using a bioanalyzer or Qubit to precisely quantify the amount and quality of DNA.

Minor comments:

1. There are quite a number of typos, which require careful correction

2. I also suggest that the authors improve the organization of their result figures and captions related to the phylogenetic analyses. (e.g., what the numbers represent? what the distinction is between “GE_Strongyloides_stercoralis_” and “Strongyloides_stercoralis” within the same phylogenetic tree??). In its current form, these figures are somewhat difficult to interpret.

**Do you want your identity to be public for this peer review?** For information about this choice, including consent withdrawal, please see our Privacy Policy

Reviewer #1: No

Reviewer #2: No

---

## [Author Response · Author response to Decision Letter 1]

28 Oct 2025

PONE-D-25-39373

Next-Gen Intestinal Parasite Detection: Leveraging Metataxonomics for Improved Diagnosis of Intestinal Protists and Helminths

Rebuttal Letter

R/: We are very grateful to the editor and reviewers for investing their time in carefully reviewing our manuscript and for their thoughtful comments, critiques, and recommendations. We have made every possible effort to improve the manuscript by addressing all the feedback, and we are confident that this revised version represents a significant improvement.

R:/ We have adjusted the manuscript format according to the journal’s recommendations and renamed the figures and supplementary materials as suggested.

“This study was funded by Escuela de Microbiología-CODI, Universidad de Antioquia, under grant code 2023-64370.”

R:/ Thank you for the comment. Indeed, the funders had no role in the study, and we have added this clarification to the Acknowledgments section:

“This study was funded by Escuela de Microbiología-CODI, Universidad de Antioquia, under grant code 2023-64370. The funders had no role in study design, data collection and analysis, decision to publish, or preparation of the manuscript.”

3. Please note that your Data Availability Statement is currently missing [the repository name and/or the DOI/accession number of each dataset OR a direct link to access each database]. If your manuscript is accepted for publication, you will be asked to provide these details on a very short timeline. We therefore suggest that you provide this information now, though we will not hold up the peer review process if you are unable.

R:/ The data were made publicly available on October 16th, and the BioProject containing the raw data can be accessed through the following link:

https://www.ncbi.nlm.nih.gov/bioproject/PRJNA1293464

This information was also added to the manuscript.

“This study was funded by Escuela de Microbiología-CODI, Universidad de Antioquia, under grant code 2023-64370.”

“This study was funded by Escuela de Microbiología-CODI, Universidad de Antioquia, under grant code 2023-64370”

R:/ Thank you for the clarification. The funding information was removed from the Acknowledgements section and is not mentioned elsewhere in the manuscript. The funding information provided in the online submission form should be as follows:

This study was funded by Escuela de Microbiología-CODI, Universidad de Antioquia, under grant code 2023-64370. The funders had no role in study design, data collection and analysis, decision to publish, or preparation of the manuscript.

5. We note that there is identifying data in the Supporting Information file <Supp_TABLE_1_NGS_Mic.xlsx>. Due to the inclusion of these potentially identifying data, we have removed this file from your file inventory. Prior to sharing human research participant data, authors should consult with an ethics committee to ensure data are shared in accordance with participant consent and all applicable local laws.

-Location data

R:/ Thanks for the clarification. The table S1 was modified to remove any information related to age, sex, or any other details that could compromise participant privacy.

R:/ Thanks for the observation. We took it into consideration while preparing the revised version of the manuscript.

Additional Editor Comments:

The claim that metataxonomics outperforms microscopy is presented too strongly. The results show that sequencing was superior for detecting Strongyloides and Blastocystis, yet microscopy was more effective for Trichuris and Ascaris. This contrast demonstrates that the performance of each method is parasite dependent. The interpretation should therefore highlight complementarity rather than positioning one method as superior to the other.

The treatment of discrepancies between methods is not sufficiently critical. For example, the lack of Giardia detections in sequencing is attributed to true absence in the population, yet microscopy found positive cases. Technical limitations such as primer mismatch or inefficient DNA extraction provide more plausible explanations. Similar issues appear for Iodamoeba and Cyclospora. The manuscript would benefit from a stronger discussion of methodological causes for these mismatches, ideally supported by targeted molecular validation.

R:/ We fully agree with the comment and have revised the manuscript to present the two methods as complementary. Our main idea is that this technology represents a new tool within the parasitologist’s technological toolbox. Additionally, we have included the following paragraph in the manuscript to further elaborate on the possible explanations for the incongruent results observed between the NGS and microscopy analyses:

The most discordant results were observed for other protozoa, namely Chilomastix, Cyclospora, Cystoisospora, and Giardia. For these protozoan taxa, there was no concordance between microscopy and NGS metataxonomic; each was detected by only one method, with no mutual confirmation. This highlights the inherent challenge of capturing the full diversity of parasites using a single methodological approach, given the varying biological and molecular characteristics of these organisms. In 18S rRNA metataxonomics, several technical factors may contribute to false-negative results. First, DNA extraction protocols may not fully disrupt the resistant cyst or oocyst walls of certain protozoa, limiting the recovery of amplifiable DNA (Naushad et al., 2025). Second, sequence polymorphisms in the target regions can affect the binding efficiency of universal primers, introducing amplification bias (Bradley et al., 2016). Third, the low abundance of parasite DNA relative to host and microbiome DNA can reduce amplification efficiency (Frau et al., 2019). Additionally, variability in ribosomal gene copy number among taxa and preferential amplification of shorter fragments, particularly when targeting V3-V4 regions, can further affect detection sensitivity (Chihi et al., 2022; Stensvold, 2024). Taken together, these factors indicate that absence of detection by NGS should be interpreted cautiously, as it may reflect technical limitations rather than true absence of the organism.

In the case of Chilomastix, bioinformatic analysis revealed an issue similar to that observed with Iodamoeba: the reverse primer target region shows poor sequence conservation in this genus, likely impairing efficient primer annealing and thus detection by PCR. For Cyclospora, although we cannot entirely rule out primer incompatibility, the same primer set has successfully amplified other apicomplexan parasites (e.g., Cystoisospora, Cryptosporidium) in past analyses. Therefore, no definitive conclusion can be made regarding Cyclospora detection.

Interestingly, Giardia was not detected by NGS in any of the samples analyzed in this study, even though we have previously reported its successful detection using the same primers and protocols in human stool samples and wastewater from Colombia (Rozo-Montoya et al., 2023). This observation is consistent with other metataxonomic studies that have reported limited or no detection of Giardia, highlighting persistent technical challenges (Ramayo-Caldas et al., 2020; Stensvold et al., 2021; Wylezich et al., 2019, 2020). Several authors have suggested that primer mismatches, low abundance, and especially inefficient DNA extraction from the parasite’s resistant cyst wall may underline these failures. Standard fecal DNA extraction protocols may be suboptimal for Giardia which often require more intensive lysis methods— such as mechanical disruption or repeated freeze-thaw cycles— to ensure adequate DNA release (Babaei et al., 2011).

The sample size of 65 is small and combines specimens from two time periods almost seven years apart. Pooling across such a wide temporal gap introduces the possibility of confounding through epidemiological changes or laboratory practice differences. The authors should address this explicitly and discuss how these potential biases were controlled or accounted for in the study design.

R:/ We thank the reviewer for raising these relevant points. While the cohort of 65 samples may appear limited, it was sufficient to capture the diversity of intestinal parasites circulating in the studied populations and to demonstrate the applicability and performance of the metataxonomic–phylogenetic framework. Our primary objective was methodological—to evaluate the capacity of NGS-based metataxonomics to detect and resolve multiple intestinal parasites within real field samples—rather than to perform a large-scale epidemiological survey. Moreover, we fully acknowledge the potential implications of combining samples collected at different time periods. Our primary objective was to perform a methodological comparison between NGS and microscopy for parasite detection in an endemic setting. The relative stability of parasitic infections in such regions provides a suitable context in which pooling these samples enables a robust evaluation of the technique’s capacity to reveal diversity that might be overlooked by microscopy alone. Moreover, it is important to note that all stool samples were processed within a short time after collection—within a few days—and DNA extraction was performed using the same commercial kit by the same highly trained Ph.D.-level researcher experienced in molecular biology techniques. All purified DNA samples were stored at −80 °C to ensure optimal preservation and minimize degradation. These steps were taken to control potential biases as much as possible. In response to this comment, we will add a paragraph to the discussion explicitly addressing how long-term sample storage could affect DNA quality and NGS performance, thereby strengthening the interpretation of our findings:

“Another potential limitation of this study is the long-term storage of DNA samples prior to NGS analysis. While prolonged preservation at -80oC generally maintains DNA integrity, factors such as storage duration and freeze-thaw cycles can influence quality. Evidence from microbiota research indicates that fecal DNA can remain viable for extended periods, even up to 14 years, under stable frozen conditions (Kia et al., 2016; Kim et al., 2023; Nel Van Zyl et al., 2020). However, parasite specific data is more limited. Although studies suggest that parasite DNA, such as from Giardia, remains detectable after long-term storage (Kuk et al., 2012), the effect on the broader parasitome is less clear. In our study, all samples were maintained under a strict and uniform storage protocol at -80oC from collection until DNA extraction. This consistent handling ensures that any potential DNA degradation would be systematic across the entire sample set. Consequently, such effects would likely reduce overall detection sensitivity uniformly, without compromising our core finding that NGS identifies greater parasite diversity than microscopy in these archival samples.”

Figures showing detection frequencies are presented as simple bar charts without any measure of uncertainty. Although presence and absence data are used, statistical analysis or variability indicators would allow readers to assess the strength of the comparisons. In the phylogenetic analyses, bootstrap values are presented as definitive, yet even high values need careful interpretation. A more cautious discussion of these analyses is necessary.

R:/ We sincerely thank the reviewer for this thoughtful and constructive observation. We fully agree that including measures of uncertainty and adopting a more cautious interpretation of the phylogenetic analyses will strengthen the manuscript.

It is important to note that our main goal in this work is to demonstrate how classical parasitological methods can be complemented by NGS-based metataxonomics and to highlight the potential limitations inherent to each approach. We believe that the results presented here will open new avenues for research in diagnostic methodologies, and that the conclusions drawn from this study will help guide the design of more quantitative and statistically robust analyses in future investigations.

Regarding the phylogenetic analyses, we have carefully revised the text to moderate our interpretation of bootstrap support values, emphasizing that—even when high—these values represent statistical confidence within the dataset and should not be regarded as absolute indicators of phylogenetic certainty. As an additional line of support, the selection of molecular operational taxonomic units (mOTUs) corresponding to each genus was validated through BLASTN comparisons against reference sequences. These comparisons consistently showed nea

---

## [Editor Report · Decision Letter 1]

2 Nov 2025

Dear Dr. Alzate,

Thank you for submitting your manuscript to PLOS ONE. After careful consideration, we feel that it has merit but does not fully meet PLOS ONE’s publication criteria as it currently stands. Therefore, we invite you to submit a revised version of the manuscript that addresses the points raised during the review process.

We look forward to receiving your revised manuscript.

Kind regards,

Petr Heneberg

Academic Editor

PLOS ONE

**Journal Requirements:**

**Additional Editor Comments:**

The revision substantially improved the paper. I have just one minor comment - the manuscript would benefit from a stronger emphasis on One Health and zoonotic perspectives, framing intestinal parasite detection within the broader context of human–animal–environment interactions. Incorporating a One Health viewpoint aligns the study with current global efforts to understand and mitigate cross-species transmission risks through integrated diagnostic and surveillance strategies. For instance, One Health (18, 2024, 100679) discusses how the control of companion animal parasites directly influences human health outcomes and advocates for coordinated One Health approaches to reduce zoonotic burdens. Similarly, Philosophical Transactions of the Royal Society B (379, 2023, 20220445) reviews the debated zoonotic potential of Strongyloides infections transmitted between dogs and humans, underscoring the clinical and epidemiological importance of molecular differentiation across hosts. A global synthesis published in PLOS Global Public Health (5, 2025, e0004614) examines cross-host soil-transmitted helminth infections in humans and domestic or livestock animals, highlighting that many intestinal parasites circulate within multi-host ecological systems rather than exclusively human reservoirs. In addition, Review One Health (21, 2025, 101166) presents recent evidence for bidirectional transmission of Blastocystis between pets and their owners and calls for standardized genomic approaches to discern true zoonotic events from shared environmental exposures. Integrating insights from these studies would allow to position metataxonomic diagnostics not merely as a technical advancement but as a practical One Health surveillance tool capable of mapping host–parasite networks, detecting emerging zoonoses, and informing cross-sectoral control strategies linking human, veterinary, and environmental health.

---

## [Author Response · Author response to Decision Letter 2]

6 Nov 2025

We thank the Editor for the insightful suggestion. Accordingly, we have included the following paragraph in the Discussion section, as recommended by the reviewer.

“Human nematode and protozoan infections often emerge from interconnected human–animal (particularly companion animal)–environment relationships (63). Integrating high-performance diagnostic approaches, such as metataxonomic sequencing, can provide a powerful means to map these host–parasite networks within complex environmental contexts. Geohelminths such as Strongyloides spp. have shown debated zoonotic potential between dogs and humans, highlighting the need for molecular differentiation to clarify cross-host transmission (64). Ascaris suum commonly infects domestic swine; however previous studies suggest that pig Ascaris suum may cause zoonotic infections in children (65). Furthermore, Blastocystis transmission appears to occur bidirectionally between pets and their owners, supporting the importance of genomic tools to identify true zoonotic events (66). Within this broader context, metataxonomic and phylogenetic frameworks such as the one applied in this study can be envisioned as practical One Health surveillance tools, enabling mapping of host–parasite networks, early detection of emerging zoonoses, and evidence-based guidance for integrated control strategies that link human, veterinary, and environmental health.”

---

## [Editor Report · Decision Letter 2]

9 Nov 2025

Dear Dr. Alzate,

Thank you for submitting your manuscript to PLOS ONE. After careful consideration, we feel that it has merit but does not fully meet PLOS ONE’s publication criteria as it currently stands. Therefore, we invite you to submit a revised version of the manuscript that addresses the points raised during the review process.

We look forward to receiving your revised manuscript.

Kind regards,

Petr Heneberg

Academic Editor

PLOS ONE

Journal Requirements:

Additional Editor Comments:

The references need to be double-checked, unnecessary Spanish words removed, and missing letters added. The suggested text was added, but it contains some weird references instead of the suggested ones. The reference on Strongyloides was replaced by some irrelevant one; similarly, the reference on Blastocystis was replaced by a paper that only marginally deals with Blastocystis itself. Remove the text likely generated by LLM (the presence of em-dashes, and unnecessary words, such as “Finally”, etc., and unusual words, such as “plausible” suggests use some AI-generated text). Add a real discussion of your results. Include a Statistics section and analyze your data using relevant statistical methods.

---

## [Author Response · Author response to Decision Letter 3]

9 Dec 2025

PONE-D-25-39373

Next-Gen Intestinal Parasite Detection: Leveraging Metataxonomics for Improved

Diagnosis of Intestinal Protists and Helminths

Rebuttal Letter

1- Journal Requirements:

Additional Editor Comments:

The references need to be double-checked, unnecessary Spanish words removed, and missing letters added. The suggested text was added, but it contains some weird references instead of the suggested ones. The reference on Strongyloides was replaced by some irrelevant one; similarly, the reference on Blastocystis was replaced by a paper that only marginally deals with Blastocystis itself. Remove the text likely generated by LLM (the presence of em-dashes, and unnecessary words, such as “Finally”, etc., and unusual words, such as “plausible” suggests use some AI-generated text). Add a real discussion of your results. Include a Statistics section and analyze your data using relevant statistical methods.

R/ We sincerely appreciate the reviewer’s thoughtful comment. In response, we have carefully revised the manuscript by reviewing and correcting all references to ensure that each cited work is relevant, accurate, and properly formatted. In addition, we removed residual Spanish words and typographical inconsistencies throughout the text. We also incorporated the suggested references to strengthen the discussion, emphasizing the relevance of metataxonomic approaches within a One Health framework. We made every effort to ensure that the discussion fully addresses the findings obtained in this study and their broader implications. The revised section now includes the following paragraph:

Intestinal protozoa and nematode infections occur through interactions between humans, their domestic animals—both companion and livestock— and the surrounding environment. Molecular studies already show parasite exchange across species barriers. Comparative analyses of Strongyloides 18S and cox1 sequences from humans and dogs have revealed two major S. stercoralis lineage, one apparently restricted to canines, and another shared among dogs, cats, humans, and non-human primates (56). These findings indicated a limited yet plausible zoonotic link, though the direction and geographic extent of transmission remain uncertain, underscoring the need for co-sampling of hosts and population genetic analyses to clarify cross-species dynamics. Human-associated Blastocystis subtypes (ST1-ST3) are frequently detected in cohabiting pets and their owners, suggesting either bidirectional transmission or exposure to shared environmental sources (57). Given that most studies are cross-sectional, current evidence provides only limited insights into parasite transmission, highlighting the need for high-resolution genotyping, longitudinal sampling, and environmental assessments. Similarly, genetic analyses show extensive overlap among soil-transmitted helminths in humans and animals, suggesting shared transmission cycles rather than exclusively human reservoirs (8). These findings indicate the need for integrated parasite control and management to mitigate zoonotic and environmental risks. Monitoring and preventive measures across veterinary, medical, and public health sectors are critical to reduce cross-species transmission, but current evidence reveals gaps in treatment, evaluation and coordinated efforts involving health and veterinary professionals, researchers, industry, and governmental authorities (58). In this context, metataxonomic approaches offer a powerful tool to advance One Health investigations, by providing high-resolution data on parasite diversity, geographic distribution, and potential reservoirs, supporting informed, evidence-based interventions across humans, animals, and the environment.

We also appreciate the reviewer’s observation regarding language and style. The manuscript has been thoroughly reviewed and edited to improve clarity and consistency with the formal scientific tone required by PLOS ONE. Unnecessary words and stylistic elements have been removed to ensure a clear and objective presentation of the findings.

A dedicated section has been added to describe the statistical analyses performed. These analyses correspond to a preliminary exploratory assessment, which we consider appropriate for the current scope of the study and sufficient to support the interpretation of the data. More advanced statistical analyses will be incorporated in future studies as additional datasets become available. Here is the paragraph included in the methodology section:

Statistical analysis

Descriptive analyses were performed to compare the detection outcomes obtained by microscopy and metataxonomic (NGS-based) approaches. The presence or absence of each parasite taxon was summarized as qualitative variables, and detection frequencies were calculated as the proportion of positive samples per method. Simple concordance rates between both diagnostic approaches were also calculated to provide a preliminary measure agreement. Results were visualized using comparative bar plots generated in R (v4.3.1) with the ggplot2 package. No inferential statistical tests were applied, as the objective of this preliminary study was to provide an exploratory, qualitative comparison of diagnostic performance. More comprehensive statistical analyses could be performed in future studies with larger sample sizes.

Furthermore, we have made a minor modification to the introductory paragraph of the Results section to clearly indicate that this work represents a preliminary comparison between traditional microscopy and metataxonomic approaches for parasite detection. Here is the modified paragraph:

This study aimed to provide a preliminary comparison of parasite detection in human fecal samples using two approaches: traditional smear microscopy performed by trained parasitologists, and a metataxonomic method targeting the V4 hypervariable region of the eukaryotic 18S rDNA. Previous tests confirmed that the primers used in this study are effective for detecting protists and nematodes but do not amplify DNA from cestodes and trematodes. As a result, these groups were excluded from the scope of our analysis. Accordingly, our study focused on the detection of human intestinal protists and nematodes.

In addition, the abstract has been revised to reflect the exploratory nature of the study, including the challenges encountered with certain taxa, the differential detection by each method, and the limitations of current primers and DNA extraction techniques. This adjustment ensures that the manuscript accurately conveys the scope and preliminary character of the study. Here is the modified abstract:

Intestinal parasites continue to pose a significant public health burden in low- and middle-income countries and are increasingly recognized in developed regions. Traditional diagnostic methods, primarily based on microscopy, remain widely used despite limitations in sensitivity and taxonomic resolution. In this exploratory study, we applied a next-generation sequencing (NGS)-based metataxonomic approach, integrated with classical phylogenetic methods, to characterize intestinal parasites in rural Colombian populations. We compare its performance with conventional microscopy, focusing on both protist and geohelminth detection. Metataxonomics outperformed microscopy in detecting Strongyloides stercoralis and enabled precise species and subtype level assignment for Blastocystis and Entamoeba spp., revealing frequent mixed infections. Microscopy detected Trichuris trichiura, Giardia, Cyclospora, and Chilomastix more effectively, highlighting limitations of current primers and DNA extraction methods. Cystoisospora was only identified by NGS. These results demonstrate the utility of NGS-based metataxonomics for broad parasite detection while emphasizing areas for methodological improvement and providing a foundation for future, larger-scale studies.

We trust that these revisions address the reviewer’s concerns and enhance the overall clarity, rigor, and scientific value of the manuscript.

---

## [Editor Report · Decision Letter 3]

10 Dec 2025

Next-Gen Intestinal Parasite Detection: Leveraging Metataxonomics for Improved Diagnosis of Intestinal Protists and Helminths

PONE-D-25-39373R3

Dear Dr. Alzate,

We’re pleased to inform you that your manuscript has been judged scientifically suitable for publication and will be formally accepted for publication once it meets all outstanding technical requirements.

Kind regards,

Petr Heneberg

Academic Editor

PLOS One
---

## [Editor Report · Acceptance letter]

PONE-D-25-39373R3

PLOS One

Dear Dr. Alzate,

I'm pleased to inform you that your manuscript has been deemed suitable for publication in PLOS One. Congratulations! Your manuscript is now being handed over to our production team.

Kind regards,

on behalf of

Dr. Petr Heneberg

Academic Editor

PLOS One